# PSTNet: Point Spatio-Temporal Convolution on Point Cloud Sequences

**Hehe Fan[1], Xin Yu[2], Yuhang Ding[3], Yi Yang[2] & Mohan Kankanhalli[1]**
[1]School of Computing, National University of Singapore
[2]ReLER, University of Technology Sydney
[3]Baidu Research

## Abstract

Point cloud sequences are irregular and unordered in the spatial dimension while exhibiting regularities and order in the temporal dimension. Therefore, existing grid based convolutions for conventional video processing cannot be directly applied to spatio-temporal modeling of raw point cloud sequences. In this paper, we propose a point spatio-temporal (PST) convolution to achieve informative representations of point cloud sequences. The proposed PST convolution first disentangles space and time in point cloud sequences. Then, a spatial convolution is employed to capture the local structure of points in the 3D space, and a temporal convolution is used to model the dynamics of the spatial regions along the time dimension. Furthermore, we incorporate the proposed PST convolution into a deep network, namely PSTNet, to extract features of point cloud sequences in a hierarchical manner. Extensive experiments on widely-used 3D action recognition and 4D semantic segmentation datasets demonstrate the effectiveness of PSTNet to model point cloud sequences.

## 1 Introduction

Modern robotic and automatic driving systems usually employ real-time depth sensors, such as LiDAR, to capture the geometric information of scenes accurately while being robust to different lighting conditions. A scene geometry is thus represented by a 3D point cloud, *i.e.*, a set of measured point coordinates $\{(x, y, z)\}$. Moreover, when RGB images are available, they are often used as additional features associated with the 3D points to enhance the discriminativeness of point clouds. However, unlike conventional grid based videos, dynamic point clouds are irregular and unordered in the spatial dimension while points are not consistent and even flow in and out over time. Therefore, existing 3D convolutions on grid based videos (Tran et al., 2015; Carreira & Zisserman, 2017; Hara et al., 2018) are not suitable to model raw point cloud sequences, as shown in Fig. 1.

To model the dynamics of point clouds, one solution is converting point clouds to a sequence of 3D voxels, and then applying 4D convolutions (Choy et al., 2019) to the voxel sequence. However, directly performing convolutions on voxel sequences require a large amount of computation. Furthermore, quantization errors are inevitable during voxelization, which may restrict applications that require precise measurement of scene geometry. Another solution MeteorNet (Liu et al., 2019e) is extending the static point cloud method PointNet++ (Qi et al., 2017b) to process raw point cloud sequences by appending 1D temporal dimension to 3D points. However, simply concatenating coordinates and time together and treating point cloud sequences as unordered 4D point sets neglect the temporal order of timestamps, which may not properly exploit the temporal information and lead to inferior performance. Moreover, the scales of spatial displacements and temporal differences in point cloud sequences may not be compatible. Treating them equally is not conducive for network optimization. Besides, MeteorNet only considers spatial neighbors and neglects the local dependencies of neighboring frames. With the use of whole sequence length as its temporal receptive field, MeteorNet cannot construct temporal hierarchy. As points are not consistent and even flow in and out of the region, especially for long sequences and fast motion embedding points in a spatially local area along an entire sequence handicaps capturing accurate local dynamics of point clouds.

In this paper, we propose a point spatio-temporal (PST) convolution to directly process raw point cloud sequences. As dynamic point clouds are spatially irregular but ordered in the temporal dimension, we

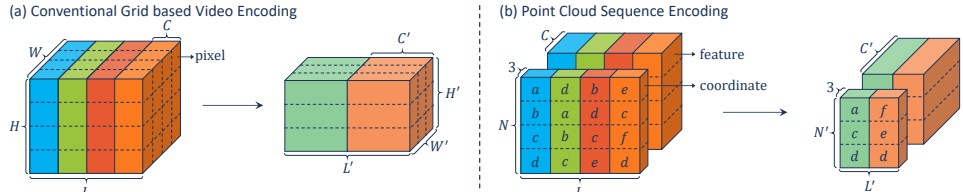

Figure 1: Illustration of grid based and point based convolutions on spatio-temporal sequences. **(a)** For a grid based video, each grid represents a feature of a pixel, where $C$, $L$, $H$ and $W$ denote the feature dimension, the number of frames, height and width, respectively. A 3D convolution encodes an input to an output of size $C' \times L' \times H' \times W'$. **(b)** A point cloud sequence consists of a coordinate part ($3 \times L \times N$) and a feature part ($C \times L \times N$), where $N$ indicates the number of points in a frame. Our PST convolution encodes an input to an output composed of a coordinate tensor ($3 \times L' \times N'$) and a feature tensor ($C' \times L' \times N'$). Usually, $L' \leq L$ and $N' \leq N$ so that networks can model point cloud sequences in a spatio-temporally hierarchical manner. Note that points in different frames are not consistent, and thus it is challenging to capture the spatio-temporal correlation.

decouple the spatial and temporal information to model point cloud sequences. Specifically, PST convolution consists of (i) a point based spatial convolution that models the spatial structure of 3D points and (ii) a temporal convolution that captures the temporal dynamics of point cloud sequences. In this fashion, PST convolution significantly facilitates the modeling of dynamic point clouds and reduces the impact of the spatial irregularity of points on temporal modeling. Because point cloud sequences emerge inconsistently across frames, it is challenging to perform convolution on them. To address this problem, we introduce a point tube to preserve the spatio-temporal local structure.

To enhance the feature extraction ability, we incorporate the proposed PST convolution into a spatio-temporally hierarchical network, namely PSTNet. Moreover, we extend our PST convolution to a transposed version to address point-level prediction tasks. Different from the convolutional version, the PST transposed convolution is designed to interpolate temporal dynamics and spatial features. Extensive experiments on widely-used 3D action recognition and 4D semantic segmentation datasets demonstrate the effectiveness of the proposed PST convolution and the superiority of PSTNet in modeling point clouds sequences. The contributions of this paper are fourfold:

- To the best of our knowledge, we are the first attempt to decompose spatial and temporal information in modeling raw point cloud sequences, and propose a generic point based convolutional operation, named PST convolution, to encode raw point cloud sequences.

- We propose a PST transposed convolution to decode raw point cloud sequences via interpolating the temporal dynamics and spatial feature for point-level prediction tasks.

- We construct convolutional neural networks based on the PST convolutions and transposed convolutions, dubbed PSTNet, to tackle sequence-level classification and point-level prediction tasks. To the best of our knowledge, our PSTNet is the first deep neural network to model raw point cloud sequences in a both spatially and temporally hierarchical manner.

- Extensive experiments on four datasets indicate that our method improves the accuracy of 3D action recognition and 4D semantic segmentation.

## 2 RELATED WORK

**Learning Representations on Grid based Videos.** Impressive progress has been made on generating compact and discriminative representations for RGB/RGBD videos due to the success of deep neural networks. For example, two-stream convolutional neural networks (Simonyan & Zisserman, 2014; Wang et al., 2016) utilize a spatial stream and an optical flow stream for video modeling. To summarize the temporal dependencies of videos, recurrent neural networks (Ng et al., 2015; Fan et al., 2018) and pooling techniques (Fan et al., 2017) are employed. In addition, by stacking multiple 2D frames into a 3D tensor, 3D convolutional neural networks (Tran et al., 2015; Carreira & Zisserman, 2017; Tran et al., 2018; Hara et al., 2018) are widely used to learn spatio-temporal representations for videos, and achieve promising performance. Besides, interpretable video or action reasoning methods (Zhuo et al., 2019; Fan et al., 2021b) are proposed by explicitly parsing changes in videos.

**Static Point Cloud Processing.** Static point cloud analysis has been widely investigated for many problems, such as classification, object part segmentation, scene semantic segmentation (Qi et al., 2017a;b; Li et al., 2018b; Wu et al., 2019; Thomas et al., 2019; Wang et al., 2019), reconstruction (Dai et al., 2017; Yu et al., 2018) and object detection (Chen et al., 2017; Qi et al., 2019). Most recent works aim to directly manipulate point sets without transforming coordinates into regular voxel grids. Since a point cloud is essentially a set of unordered points and invariant to permutations of its points, static point cloud processing methods mainly focus on designing effective point based spatial correlation operations that do not rely on point orderings.

**Dynamic Point Cloud Modeling.** Compared with static point cloud processing, dynamic point cloud modeling is a fairly new task. Fast and Furious (FaF) (Luo et al., 2018) converts a point cloud frame into a bird's view voxel and then extracts features via 3D convolutions. MinkowskiNet (Choy et al., 2019) converts a point cloud sequence into a 4D occupancy grid and then applies 4D Spatio-Temporal ConvNets. PointRNN (Fan & Yang, 2019) leverages point based recurrent neural networks for raw point cloud sequence forecasting. MeteorNet (Liu et al., 2019e) extends 3D points to 4D points and then appends a temporal dimension to PointNet++ to process these 4D points. 3DV (Wang et al., 2020) first integrates 3D motion information into a regular compact voxel set and then applies PointNet++ to extract representations from the set for 3D action recognition. P4Transformer (Fan et al., 2021a) employs transformer to avoid point tracking for raw point cloud sequence modeling. Niemeyer et al. (2019) learned a temporal and spatial vector field for 4D reconstruction. Prantl et al. (2020) learned stable and temporal coherent feature spaces for point based super-resolution. CaSPR (Rempe et al., 2020) learns to encode spatio-temporal changes in object shape from point clouds for reconstruction and camera pose estimation. In this paper, we propose a point based convolution to model raw point cloud sequences in a spatio-temporally hierarchical manner.

## 3 PROPOSED POINT SPATIO-TEMPORAL CONVOLUTIONAL NETWORK

In this section, we first briefly review the grid based 3D convolution operation as our point spatio-temporal (PST) convolution is motivated by it. Then, we introduce how the proposed PST convolution extracts features from dynamic point clouds in detail. In order to address dense point prediction tasks, *i.e.*, semantic segmentation, we develop a PST transposed convolution. Finally, we incorporate our operations into deep hierarchical networks to address different dynamic point cloud tasks.

### 3.1 MOTIVATION FROM GRID BASED 3D CONVOLUTION

The power of convolutional neural network (CNNs) comes from local structure modeling by convolutions and global representation learning via hierarchical architectures. Given an input feature map $\mathbf{F} \in \mathbb{R}^{C \times L \times H \times W}$, where $C$, $L$, $H$ and $W$ denote the input feature dimension, length, height and width, 3D convolution is designed to capture the spatio-temporal local structure of $\mathbf{F}$, written as:

$$\mathbf{F}_t'^{(x,y)} = \sum_{k=-\lfloor l/2 \rfloor}^{\lfloor l/2 \rfloor} \sum_{i=-\lfloor h/2 \rfloor}^{\lfloor h/2 \rfloor} \sum_{j=-\lfloor w/2 \rfloor}^{\lfloor w/2 \rfloor} \mathbf{W}_k^{(i,j)} \cdot \mathbf{F}_{t+k}^{(x+i,y+j)}, \tag{1}$$

where $\mathbf{W} \in \mathbb{R}^{C' \times C \times l \times h \times w}$ is the convolution kernel, $(l, h, w)$ is the kernel size, $C'$ represents the output feature dimension and $\cdot$ is matrix multiplication. The $\mathbf{W}_k^{(i,j)} \in \mathbb{R}^{C' \times C}$ denotes the weight at kernel position $(k, i, j)$ and $\mathbf{F}_{t+k}^{(x+i,y+j)} \in \mathbb{R}^{C \times 1}$ denotes the feature of pixel at input position $(t+k, x+i, y+j)$. Usually, CNNs employ small kernel sizes, *e.g.*, $(3, 3, 3)$, which are much smaller than the input size, to effectively model the local structure. To construct hierarchical CNNs, stride $(> 1)$ is used to subsample input feature maps during the convolution operation. In this fashion, following convolutional layers will have relatively larger receptive fields.

### 3.2 POINT SPATIO-TEMPORAL (PST) CONVOLUTION

Let $\boldsymbol{P}_t \in \mathbb{R}^{3 \times N}$ and $\boldsymbol{F}_t \in \mathbb{R}^{C \times N}$ denote the point coordinates and features of the $t$-th frame in a point cloud sequence, where $N$ and $C$ denote the number of points and feature channels, respectively. Given a point cloud sequence $([\boldsymbol{P}_1; \boldsymbol{F}_1], [\boldsymbol{P}_2; \boldsymbol{F}_2], \cdots, [\boldsymbol{P}_L; \boldsymbol{F}_L])$, the proposed PST convolution will encode the sequence to $([\boldsymbol{P}_1'; \boldsymbol{F}_1'], [\boldsymbol{P}_2'; \boldsymbol{F}_2'], \cdots, [\boldsymbol{P}_{L'}'; \boldsymbol{F}_{L'}'])$, where $L$ and $L'$ indicate the number of frames, and $\boldsymbol{P}_t' \in \mathbb{R}^{3 \times N'}$ and $\boldsymbol{F}_t' \in \mathbb{R}^{C' \times N'}$ represent the encoded coordinates and features.

### 3.2.1 DECOMPOSING SPACE AND TIME IN POINT CLOUD SEQUENCE MODELING

Because point clouds are irregular and unordered, grid based 3D convolution cannot be directly applied to point cloud sequences. As point cloud sequences are spatially irregular and unordered but temporally ordered, this motivates us to decouple these two dimensions in order to reduce the impacts of the spatial irregularity of points on temporal modeling. Moreover, the scales of spatial displacements and temporal differences in point cloud sequences may not be compatible. Treating them equally is not conducive for network optimization. By decoupling the spatio-temporal information in point cloud sequences, we not only make the spatio-temporal modeling easier but also significantly improve the ability to capture the temporal information. Therefore, our PST convolution is formulated as:

$$
\begin{aligned}
\boldsymbol{F}_t^{\prime(x,y,z)} &= \sum_{k=-\lfloor l/2 \rfloor}^{\lfloor l/2 \rfloor} \sum_{\|(\delta_x,\delta_y,\delta_z)\| \leq r} \mathbf{W}_k^{(\delta_x,\delta_y,\delta_z)} \cdot \boldsymbol{F}_{t+k}^{(x+\delta_x,y+\delta_y,z+\delta_z)}, \\
&= \sum_{k=-\lfloor l/2 \rfloor}^{\lfloor l/2 \rfloor} \sum_{\|(\delta_x,\delta_y,\delta_z)\| \leq r} \mathbf{T}_k^{(\delta_x,\delta_y,\delta_z)} \cdot \left( \mathbf{S}_k^{(\delta_x,\delta_y,\delta_z)} \cdot \boldsymbol{F}_{t+k}^{(x+\delta_x,y+\delta_y,z+\delta_z)} \right),
\end{aligned}
\tag{2}
$$

where $(x,y,z) \in \boldsymbol{P}_t$, $(\delta_x, \delta_y, \delta_z)$ represents displacement and $r$ is spatial search radius. The convolution kernel $\mathbf{W} \in \mathbb{R}^{C' \times C \times l} \times \mathbb{R}$ is decomposed into a spatial convolution kernel $\mathbf{S} \in \mathbb{R}^{C_m \times C \times l} \times \mathbb{R}$ and a temporal convolution kernel $\mathbf{T} \in \mathbb{R}^{C' \times C_m \times l} \times \mathbb{R}$ and $C_m$ is the dimension of the intermediate feature. Moreover, because space and time are orthogonal and independent of each other, we further decompose spatial and temporal modeling as:

$$
\boldsymbol{F}_t^{\prime(x,y,z)} = \sum_{k=-\lfloor l/2 \rfloor}^{\lfloor l/2 \rfloor} \mathbf{T}_k \cdot \sum_{\|(\delta_x,\delta_y,\delta_z)\| \leq r} \mathbf{S}^{(\delta_x,\delta_y,\delta_z)} \cdot \boldsymbol{F}_{t+k}^{(x+\delta_x,y+\delta_y,z+\delta_z)},
\tag{3}
$$

where $\mathbf{S} \in \mathbb{R}^{C_m \times C} \times \mathbb{R}$ and $\mathbf{T} \in \mathbb{R}^{C' \times C_m \times l}$. The above decomposition can also be expressed as applying temporal convolution first and then spatial convolution to input features. However, doing so requires point tracking to capture point motion. Because it is difficult to achieve accurate point trajectory and tracking points usually relies on point colors and may fail to handle colorless point clouds, we opt to model the spatial structure of irregular points first, and then capture the temporal information from the spatial regions. In this fashion, Eq. (3) can be rewritten as:

$$
\boldsymbol{M}_t^{(x,y,z)} = \sum_{\|(\delta_x,\delta_y,\delta_z)\| \leq r} \mathbf{S}^{(\delta_x,\delta_y,\delta_z)} \cdot \boldsymbol{F}_t^{(x+\delta_x,y+\delta_y,z+\delta_z)}, \quad \boldsymbol{F}_t^{\prime(x,y,z)} = \sum_{k=-\lfloor l/2 \rfloor}^{\lfloor l/2 \rfloor} \mathbf{T}_k \cdot \boldsymbol{M}_{t+k}^{(x,y,z)}.
\tag{4}
$$

The spatial convolution captures spatial local structures of point clouds in 3D space, while the temporal convolution aims to describe the temporal local dynamics of point cloud sequences along the time dimension. In order to capture the distributions of neighboring points, it is required to learn a convolution kernel $\mathbf{S}$ for all displacements. However, this is impossible because point displacements are not discrete. To address this problem, we convert the kernel to a function of displacements, defined as:

$$
\boldsymbol{M}_t^{(x,y,z)} = \sum_{\|(\delta_x,\delta_y,\delta_z)\| \leq r} f\left((\delta_x,\delta_y,\delta_z); \boldsymbol{\theta}\right) \cdot \boldsymbol{F}_t^{(x+\delta_x,y+\delta_y,z+\delta_z)},
\tag{5}
$$

where $f : \mathbb{R}^{1\times 3} \to \mathbb{R}^{C_m \times C}$ is a function of $(\delta_x, \delta_y, \delta_z)$ parametrized by $\boldsymbol{\theta}$ to generate different $\mathbb{R}^{C_m \times C}$ according to different displacements. There can be many ways to implement the $f$ function. In principle, the $f$ function should be both computation-efficient and memory-efficient so that PST convolution is able to encode long sequences. Existing static point cloud convolution methods (Li et al., 2018b; Wu et al., 2019; Wang et al., 2019) usually contains multilayer perceptrons (MLPs), which are computation and memory consuming. In this paper, we design a lightweight implementation for $f$, which contains only a few parameters. Specifically, we further decompose the $f$ function as $f\left((\delta_x, \delta_y, \delta_z); \boldsymbol{\theta}\right) = \boldsymbol{\theta}_d \cdot (\delta_x, \delta_y, \delta_z)^T \cdot \mathbf{1} \odot \boldsymbol{\theta}_s$, where $\boldsymbol{\theta} = [\boldsymbol{\theta}_d, \boldsymbol{\theta}_s]$, $\boldsymbol{\theta}_d \in \mathbb{R}^{C_m \times 3}$ is a displacement transform kernel, $\boldsymbol{\theta}_s \in \mathbb{R}^{C_m \times C}$ is a sharing kernel, $\mathbf{1} = (1, \cdots, 1) \in \mathbb{R}^{1 \times C}$ is for broadcasting, and $\odot$ is element-wise product. The sharing kernel $\boldsymbol{\theta}_s$ is to increase point feature dimension to improve the feature representation ability. The displacement kernel $\boldsymbol{\theta}_d$ is to capture spatial local structure based on displacements. In this fashion, $f$ generates a unique spatial kernel for each displacement so that the lightweight spatial convolution is able to increase feature dimension while capturing spatial local structure like conventional convolutions.

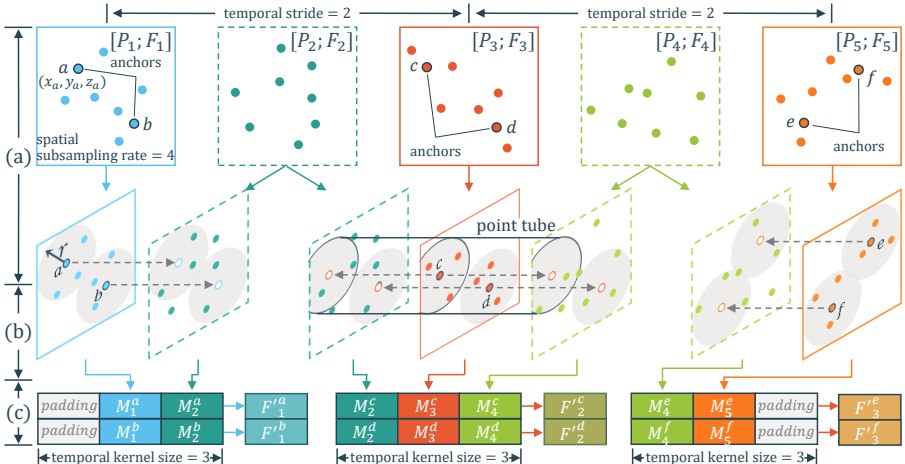

Figure 2: Illustration of the proposed point spatio-temporal (PST) convolution. The input contains $L = 5$ frames, with $N = 8$ points per frame. **(a)** Based on the temporal kernel size $l = 3$, temporal stride $s_t = 2$, and temporal padding $p = 1$, the 1st, 3rd, 5th frames are selected as temporal anchor frames. According to a spatial subsampling rate $s_s = 4$, 2 spatial anchor points are sampled by FPS in each anchor frame. The sampled anchor points are then transferred to the $\lfloor \frac{l}{2} \rfloor = 1$ nearest neighboring frames. A point tube is constructed with a spatial search radius $r$ for the anchor points. **(b)** The spatial convolution encodes the local structure around each anchor point. **(c)** The temporal convolution encodes the $l$ spatial features to a spatio-temporal feature. The original $L \times N = 5 \times 8$ sequence is encoded as a $L' \times N' = 3 \times 2$ sequence.

### 3.2.2 POINT TUBE

Grid based 3D convolutions can be easily performed on regular conventional videos by sliding along the length, height and width. Because point cloud sequences are irregular and unordered in 3D space and emerge inconsistently across different frames, it is challenging to perform convolution on them. To address this problem, we introduce a point tube to preserve the spatio-temporal local structure. In contrast to pixel cubes in 3D convolution, in which pixels are distributed regularly, point tubes are dynamically generated based on the input sequences so that dense areas have more tubes than sparse ones. Specifically, the point tube is constructed as follows:

**Temporal anchor frame selection.** For an entire sequence of point clouds, we need to select some anchor frames to generate our tubes. Temporal anchor frames in a point cloud sequence are automatically selected based on temporal kernel size ($l$), temporal stride ($s_t$) and temporal padding ($p$), where $l$ is set to an odd number so that an anchor frame is located in the middle of a point tube. Moreover, we set $\lfloor \frac{l}{2} \rfloor \geq p$ to avoid selecting a padding frame as an anchor frame.

**Spatial anchor point sampling.** Once a temporal anchor frame is selected, we need to choose spatial anchor points that can represent the distribution of all the points in the frame. Given a spatial subsampling rate $s_s$, this operation aims to subsample $N$ points to $N' = \lfloor \frac{N}{s_s} \rfloor$ points. We employ the farthest point sampling (FPS) (Qi et al., 2017b) to sample points in each anchor frame. Point tubes are generated according to sampled anchor points.

**Transferring spatial anchor points.** 3D convolutions can effectively capture local changes within a cube of a kernel size $(l, h, w)$ rather than tracking a specific pixel. Following this idea, we propagate the positions of sampled anchor points to the neighboring frames without tracking, and they are regarded as the anchor points in those frames. Specifically, each anchor point is transferred to the $\lfloor \frac{l}{2} \rfloor$ nearest frames. The original and transferred anchor points form the central axis of a tube.

**Spatial neighbor.** This step aims to find the spatial neighbors of every anchor point in each frame for performing spatial convolution. As indicated in Eq. (5), radius neighborhood $r$ is used to search neighbors within the tube slice, where local structure of points is depicted. Note that, padding is usually used in grid based convolution to align feature maps. However, our point based spatial convolution is not conducted on grids and thus spatial padding is not employed.

By performing PST convolution on point tubes, our network is able to capture the dynamic changes in local areas. The temporal kernel size $l$ and spatial search radius $r$ allow our PST convolution to capture temporal and spatial local structure, respectively. Frame subsampling (according to $s_t$) and point subsampling (according to $s_s$) make our PSTNet be both temporally and spatially hierarchical. Global movement can be summarized by merging the information from these tubes in a spatio-temporally hierarchical manner. An illustration of PST convolution is shown in Fig. 2.

## 3.3 Point Spatio-Temporal Transposed Convolution

After PST convolution, the original point cloud sequence is both spatially and temporally subsampled. However, for point-level prediction tasks, we need to provide point features for all the original points. Thus, we develop a PST transposed convolution. Suppose $\big([\boldsymbol{P}_1'; \boldsymbol{F}_1'], [\boldsymbol{P}_2'; \boldsymbol{F}_2'], \cdots, [\boldsymbol{P}_{L'}'; \boldsymbol{F}_{L'}']\big)$ is the encoded sequence of the original one $\big([\boldsymbol{P}_1; \boldsymbol{F}_1], [\boldsymbol{P}_2; \boldsymbol{F}_2], \cdots, [\boldsymbol{P}_L; \boldsymbol{F}_L]\big)$. PST transposed convolution propagates features $(\boldsymbol{F}_1', \boldsymbol{F}_2', \cdots, \boldsymbol{F}_{L'}')$ to the original point coordinates $(\boldsymbol{P}_1, \boldsymbol{P}_2, \cdots, \boldsymbol{P}_L)$, thus outputting new features $(\boldsymbol{F}_1'', \boldsymbol{F}_2'', \cdots, \boldsymbol{F}_L'')$, where $\boldsymbol{F}_t'' \in \mathbb{R}^{C'' \times N}$ and $C''$ denotes the new feature dimension. To this end, PST transposed convolution first recovers the temporal length by a temporal transposed convolution:

$$\mathsf{T}' \cdot \boldsymbol{F}_t'^{(x,y,z)} = \big[\boldsymbol{M}_{t-\lfloor l/2 \rfloor}'^{(x,y,z)}, \cdots, \boldsymbol{M}_{t+\lfloor l/2 \rfloor}'^{(x,y,z)}\big], \tag{6}$$

where $\mathsf{T}' \in \mathbb{R}^{l \times C_m' \times C'}$ is the temporal transposed convolution kernel, and then increases the number of points by assigning temporal features to original points. Inspired by (Qi et al., 2017b), the interpolated features are weighted by inverse distance between an original point and neighboring anchor points:

$$\boldsymbol{F}_t''^{(x,y,z)} = \boldsymbol{S}' \cdot \frac{\sum_{\|(\delta_x, \delta_y, \delta_z)\| \le r} w(\delta_x, \delta_y, \delta_z) \boldsymbol{M}_t'^{(x+\delta_x, y+\delta_y, z+\delta_z)}}{\sum_{\|(\delta_x, \delta_y, \delta_z)\| \le r} w(\delta_x, \delta_y, \delta_z)}, w(\delta_x, \delta_y, \delta_z) = \frac{1}{\|(\delta_x, \delta_y, \delta_z)\|^2}, \tag{7}$$

where $\boldsymbol{S}' \in \mathbb{R}^{C'' \times C_m'}$ is a sharing kernel to enhance the interpolated features and change the feature dimension from $C_m'$ to $C''$. An illustration of PST transposed convolution is shown in Appendix A.

## 3.4 PSTNet Architectures

**PSTNet for 3D action recognition.** We employ 6 PST convolution layers and a fully-connected (FC) layer for 3D action recognition. In the 1st, 2nd, 4th, 6th layers, the spatial subsampling rate is set to 2 to halve the spatial resolution. Because point clouds are irregular and unordered, spatial receptive fields cannot be relatively enlarged by spatial subsampling. To address this problem, we progressively increase the spatial search radius. In the 2nd and 4th layers, the temporal stride is set to 2 to halve the temporal resolution. In the 2-4 layers, the temporal kernel size is set to 3 to capture temporal correlation. Temporal paddings are added when necessary. After the convolution layers, average and max poolings are used for spatial pooling and temporal pooling, respectively. Finally, the FC layer maps the global feature to action predictions.

**PSTNet for 4D semantic segmentation.** We use 4 PST convolution layers and 4 PST transposed convolution layers for 4D semantic segmentation. The spatial subsampling rate is set to 4 to reduce the spatial resolution in the 1-3 PST convolutions and 2 in the fourth PST convolution. Similar to 3D action recognition, the spatial search radius progressively increases to grow spatial receptive fields along the network layers. In the 3rd PST convolution and 2nd transposed convolution, the temporal kernel size is set to 3 to reduce and increase temporal dimension, respectively. Skip connections are added between the corresponding convolution layers and transposed convolution layers. After the last PST transposed convolution layer, a 1D convolution layer is appended for semantic predictions.

In addition, batch normalization and ReLU activation are inserted between PST convolution and transposed convolution layers. Our PSTNet architectures are illustrated in Appendix B.

## 4 Experiments

## 4.1 3D Action Recognition

To show the effectiveness in sequence-level classification, we apply PSTNet to 3D action recognition. Following (Liu et al., 2019e; Wang et al., 2020), we sample 2,048 points for each frame. Point

Table 1: Action recognition accuracy (%) on the MSR-Action3D dataset.

| Method | Input | # Frames | Accuracy |
|---|---|---|---|
| Vieira *et al.* (Vieira et al., 2012) | depth | 20 | 78.20 |
| Kläser *et al.* (Kläser et al., 2008) | depth | 18 | 81.43 |
| Actionlet (Wang et al., 2012) | skeleton | all | 88.21 |
| PointNet++ (Qi et al., 2017b) | point | 1 | 61.61 |
| MeteorNet (Liu et al., 2019e) | point | 4 | 78.11 |
| | | 8 | 81.14 |
| | | 12 | 86.53 |
| | | 16 | 88.21 |
| | | 24 | 88.50 |
| PSTNet (ours) | point | 4 | 81.14 |
| | | 8 | 83.50 |
| | | 12 | 87.88 |
| | | 16 | 89.90 |
| | | 24 | **91.20** |

Figure 3: Influence of temporal kernel size and (initial) spatial search radius on MSR-Action3D with 24 frames.

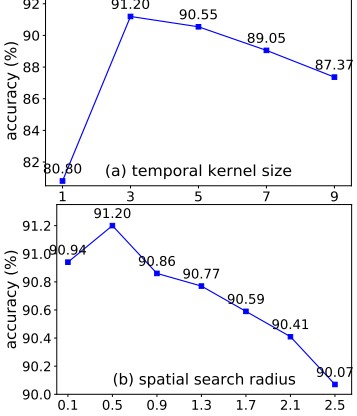

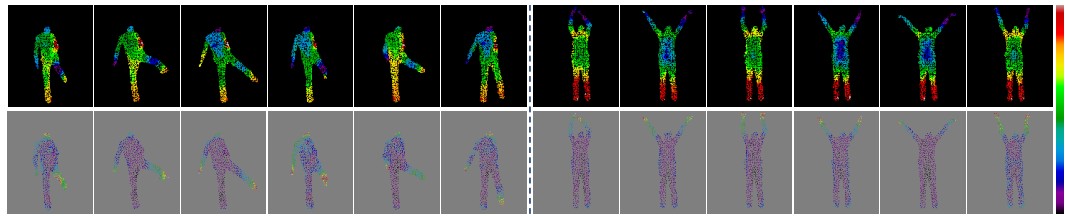

Figure 4: Visualization of PST convolution's output. Top: input point cloud sequences, where color encodes depth. Bottom: output of PST convolution, where brighter color indicates higher activation. Interestingly, PST convolution outputs high activation to salient motion. Best viewed in color.

cloud sequences are split into multiple clips (with a fixed number of frames) as inputs. For training, sequence-level labels are used as clip-level labels. For evaluation, the mean of the clip-level predicted probabilities is used as the sequence-level prediction. Point colors are not used.

### 4.1.1 MSR-ACTION3D

The MSR-Action3D (Li et al., 2010) dataset consists of 567 Kinect depth videos, with 20 action categories and 23K frames in total. We use the same training/test split as previous works (Wang et al., 2012; Liu et al., 2019e). We conduct experiments with 10 times and report the mean.

**Comparison with the state-of-the-art.** We compare our PSTNet with skeleton-based, depth-based and point-based 3D action recognition methods on this dataset. As shown in Table 1, the proposed PSTNet significantly outperforms all the state-of-the-art methods, demonstrating the superiority of our PST convolution on feature extraction. Moreover, when encoding long point sequences, our spatio-temporally hierarchical PSTNet is more effective than the spatially hierarchical MeteorNet. Specifically, from 16 frames to 24 frames, MeteorNet only achieves a slight improvement of 0.29%, while our method increases the accuracy by 1.30%.

**What does PST convolution learn?** To investigate what PST convolution learns, we visualize the output of the middle layer (*i.e.*, the 3rd layer) in Fig. 4. As ReLU is employed as the activation function, larger values indicate higher activation. As expected, PSTNet outputs higher activation on moving areas, demonstrating that our PSTNet captures the most informative clues in action reasoning.

### 4.1.2 NTU RGB+D 60 AND NTU RGB+D 120

The NTU RGB+D 60 (Shahroudy et al., 2016) is the second largest dataset for 3D action recognition. It consists of 56K videos, with 60 action categories and 4M frames in total. The videos are captured

Table 2: Action recognition accuracy (%) on the NTU RGB+D 60 and NTU RGB+D 120 datasets.

| Method | Input | NTU RGB+D 60 | | NTU RGB+D 120 | |
| --- | --- | --- | --- | --- | --- |
| | | Subject | View | Subject | Setup |
| SkeleMotion (Caetano et al., 2019) | skeleton | 69.6 | 80.1 | 67.7 | 66.9 |
| GCA-LSTM (Liu et al., 2017) | skeleton | 74.4 | 82.8 | 58.3 | 59.3 |
| FSNet (Liu et al., 2019b) | skeleton | - | - | 59.9 | 62.4 |
| Two Stream Attention LSTM (Liu et al., 2018) | skeleton | 77.1 | 85.1 | 61.2 | 63.3 |
| Body Pose Evolution Map (Liu & Yuan, 2018) | skeleton | - | - | 64.6 | 66.9 |
| AGC-LSTM (Si et al., 2019) | skeleton | 89.2 | 95.0 | - | - |
| AS-GCN (Li et al., 2019) | skeleton | 86.8 | 94.2 | - | - |
| VA-fusion (Zhang et al., 2019) | skeleton | 89.4 | 95.0 | - | - |
| 2s-AGCN (Shi et al., 2019b) | skeleton | 88.5 | 95.1 | - | - |
| DGNN (Shi et al., 2019a) | skeleton | 89.9 | 96.1 | - | - |
| HON4D (Oreifej & Liu, 2013) | depth | 30.6 | 7.3 | - | - |
| SNV (Yang & Tian, 2014) | depth | 31.8 | 13.6 | - | - |
| HOG$^2$ (Ohn-Bar & Trivedi, 2013) | depth | 32.2 | 22.3 | - | - |
| Li et al. (2018a) | depth | 68.1 | 83.4 | - | - |
| Wang et al. (2018) | depth | 87.1 | 84.2 | - | - |
| MVDI (Xiao et al., 2019) | depth | 84.6 | 87.3 | - | - |
| NTU RGB+D 120 Baseline (Liu et al., 2019a) | depth | - | - | 48.7 | 40.1 |
| PointNet++ (appearance) (Qi et al., 2017b) | point | 80.1 | 85.1 | 72.1 | 79.4 |
| 3DV (motion) (Wang et al., 2020) | voxel | 84.5 | 95.4 | 76.9 | 92.5 |
| 3DV-PointNet++ (Wang et al., 2020) | voxel + point | 88.8 | 96.3 | 82.4 | 93.5 |
| PSTNet (ours) | point | **90.5** | **96.5** | **87.0** | **93.8** |

using Kinect v2, with 3 cameras and 40 subjects (performers). The dataset defines two types of evaluation, *i.e.*, cross-subject and cross-view. The cross-subject evaluation splits the 40 performers into training and test groups. Each group consists of 20 performers. The cross-view evaluation uses all the samples from camera 1 for testing and samples from cameras 2 and 3 for training.

The NTU RGB+D 120 (Liu et al., 2019a) dataset, the largest dataset for 3D action recognition, is an extension of NTU RGB+D 60. It consists of 114K videos, with 120 action categories and 8M frames in total. The videos are captured with 106 performers and 32 collection setups (locations and backgrounds). Besides cross-subject evaluation, the dataset defines a new evaluation setting, *i.e.*, cross-setup, where 16 setups are used for training, and the others are used for testing.

**Comparison with state-of-the-art methods.** As indicated in Table 2, PSTNet outperforms all the other approaches in all evaluation settings. Particularly, as indicated by the cross-setup evaluation on NTU RGB+D 120, PSTNet outperforms the second best 3DV-PointNet++ (Wang et al., 2020) by 4.6%. Moreover, compared to 3DV that extracts motion from voxels, PSTNet directly models the dynamic information of raw point cloud sequences and thus is efficient.

**Computational efficiency.** We provide a running time comparison with the second best 3DV-PointNet++ (Wang et al., 2020). The average running time per video is shown in Fig. 5. Experiments are conducted using 1 Nvidia RTX 2080Ti GPU on NTU RGB+D 60. Impressively, compared to 3DV-PointNet++, PSTNet reduces running time by about 2s, showing that PSTNet is efficient.

## 4.2 4D Semantic Segmentation

To demonstrate that our PST convolution can be generalized to point-level dense prediction tasks, we apply PSTNet to 4D semantic segmentation. Following the works (Choy et al., 2019; Liu et al., 2019e), we conduct experiments on video clips with length of 3 frames. Note that, although semantic segmentation can be achieved from a single frame, exploring temporal consistency would facilitate exploring the structure of scenes, thus improving segmentation accuracy and robustness to noise. The widely-used standard mean Intersection over Union (mIoU) is adopted as the evaluation metric.

Synthia 4D (Choy et al., 2019) uses the Synthia dataset (Ros et al., 2016) to create 3D video sequences. The Synthia 4D dataset includes 6 sequences of driving scenarios. Each sequence consists of 4 stereo

Table 3: Semantic segmentation results on the Synthia 4D dataset.

| Method | Input | #Frms | #Params (M) | mIoU (%) |
|---|---|---|---|---|
| 3D MinkNet14 (Choy et al., 2019) | voxel | 1 | 19.31 | 76.24 |
| 4D MinkNet14 (Choy et al., 2019) | voxel | 3 | 23.72 | 77.46 |
| PointNet++ (Qi et al., 2017b) | point | 1 | 0.88 | 79.35 |
| MeteorNet (Liu et al., 2019e) | point | 3 | 1.78 | 81.80 |
| PSTNet ($l = 1$) | point | 3 | 1.42 | 80.79 |
| PSTNet ($l = 3$) | point | 3 | 1.67 | **82.24** |

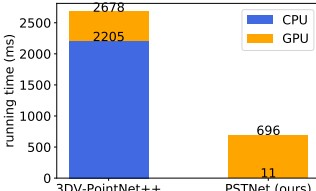

Figure 5: Comparison of recognition running time per video on NTU RGB+D 60.

RGB-D images taken from the top of a moving car. Following (Liu et al., 2019e), we use the same training/validation/test split, with 19,888/815/1,886 frames, respectively.

As seen in Table 3, PSTNet ($l = 3$) exploits the temporal information and outperforms the state-of-the-art. Moreover, our method saves 0.11M, relative 6% of parameters compared to the second best method MeteorNet (Liu et al., 2019e). We visualize a few segmentation examples in Appendix N.

### 4.3 ABLATION STUDY

**Clip length.** Usually, information is not equally distributed in sequences along time. Short point cloud clips may miss key frames and thus confuse the models as noise. Therefore, as shown in Table 1, increasing clip length (*i.e.*, the number of frames) benefits models for action recognition.

**Temporal kernel size.** The temporal kernel size $l$ controls the temporal dynamics modeling of point cloud sequences. Fig. 3(a) shows the accuracy on MSR-Action3D with different $l$. (a) When $l$ is set to 1, temporal correlation is not captured. However, PSTNet can still observe 24 frames and the pooling operation allows PSTNet to capture the pose information of an entire clip. Moreover, some actions (*e.g.*, "golf swing") can be easily recognized by a certain pose, and thus PSTNet with $l = 1$ can still achieve satisfied accuracy. (b) When $l$ is greater than 1, PSTNet models temporal dynamics and therefore improves accuracy on actions that rely on motion or trajectory reasoning (*e.g.*, "draw x", "draw tick" and "draw circle"). (c) When $l$ is greater than 3, the accuracy decreases. This mainly depends on motion in sequences. Because most actions in MSR-Action3D are fast (*e.g.*, "high arm wave"), using smaller temporal kernel size facilitates capturing fast motion, and long-range temporal dependencies will be captured in high-level layers. Since we aim to present generic point based convolution operations, we do not tune the kernel size for each action but use the same size. When PSTNet is used in 4D semantic segmentation, we observe that PSTNet ($l = 3$) improves mIoU by 1.45% compared to PSTNet ($l = 1$) that neglects temporal structure (shown in Table 3).

**Spatial search radius.** The spatial search radius $r$ controls the range of the spatial structure to be modeled. As shown in Fig. 3(b), using too small $r$ cannot capture sufficient structure information while using large $r$ will decrease the discriminativeness of spatial local structure for modeling.

## 5 CONCLUSION

In this paper, we propose an point spatio-temporal (PST) convolution to learn informative representations from raw point cloud sequences and a PST transposed convolution for point-level dense prediction tasks. We demonstrate that, by incorporating the proposed convolutions into deep networks, dubbed PSTNets, our method is competent to address various point-based tasks. Extensive experiments demonstrate that our PSTNet significantly improves the 3D action recognition and 4D semantic segmentation performance by effectively modeling point cloud sequences.

### ACKNOWLEDGMENTS

This research is supported by the Agency for Science, Technology and Research (A*STAR) under its AME Programmatic Funding Scheme (#A18A2b0046).

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

## A  PST TRANSPOSED CONVOLUTION

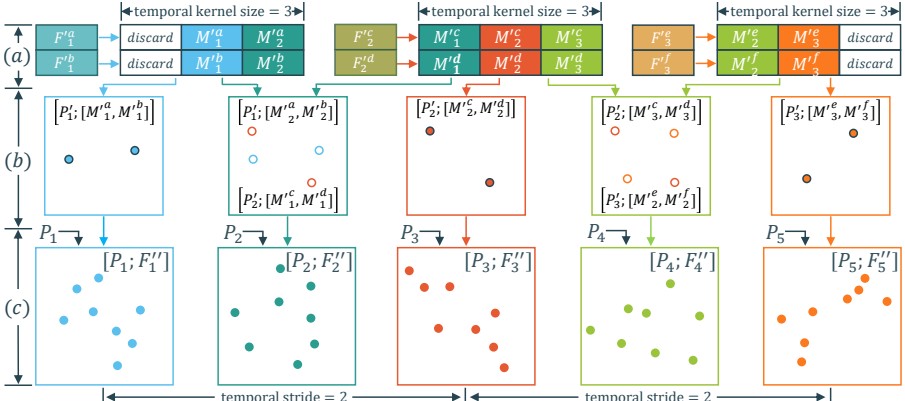

Figure 6: Illustration of the proposed PST transposed convolution. The input contains $L' = 3$ frames, with $N' = 2$ points per frame. The PST transposed convolution is to generate new original point features for the original point cloud sequence, which contains $L = 5$ frames, with $N = 8$ points per frame. The temporal kernel size $l$ is 3 and the temporal stride is $s_t$ is 2. (a) Temporal transposed convolution. (b) Temporal interpolation. (c) Spatial interpolation.

We illustrate an example of PST transposed convolution in Fig. 6. Given a convolved sequence $\left([\boldsymbol{P}_1'; \boldsymbol{F}_1'], [\boldsymbol{P}_2'; \boldsymbol{F}_2'], [\boldsymbol{P}_{3'}'; \boldsymbol{F}_{3'}']\right)$, and its original coordinate sequence $\left(\boldsymbol{P}_1, \boldsymbol{P}_2, \boldsymbol{P}_3, \boldsymbol{P}_4, \boldsymbol{P}_5\right)$, PST transposed convolution aims to generate new features $\left(\boldsymbol{F}_1'', \boldsymbol{F}_2'', \boldsymbol{F}_3'', \boldsymbol{F}_4'', \boldsymbol{F}_5''\right)$ according to the original point coordinates. First, with the temporal kernel size $l = 3$, each input point feature is decoded to 3 features by a temporal transposed convolution. Second, 5 frames are interpolated in accordance with a temporal stride $s_t = 2$, and the input points with the decoded features are assigned to interpolated frames. In this way, the temporal cross-correlation is constructed. Third, given the original coordinates of a point cloud frame (*e.g.*, $\boldsymbol{P}_1$) and the corresponding interpolated frame (*i.e.*, $\left[\boldsymbol{P}_1'; [\boldsymbol{M}_1'^a, \boldsymbol{M}_1'^b]\right]$), we generate features (*i.e.*, $\boldsymbol{F}_1''$) for all the original point coordinates.

## B  PSTNET ARCHITECTURES

### B.1  PSTNET FOR 3D ACTION RECOGNITION

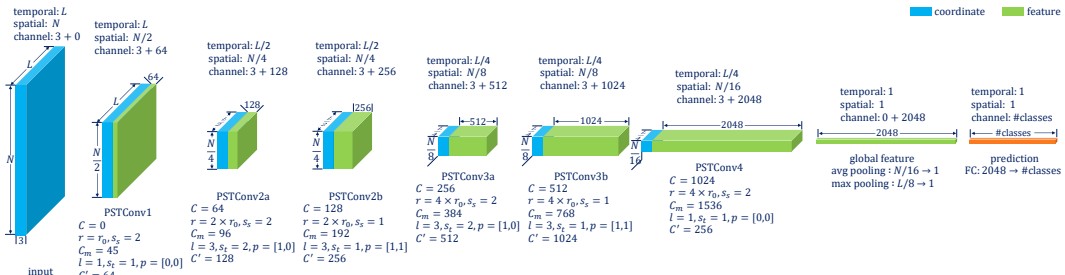

Figure 7: Hierarchical PSTNet architecture for 3D action recognition.

As shown in Fig. 7, the architecture of PSTNet for 3D action recognition consists of 6 PST convolutions, *i.e.*, PSTConv1, PSTConv2a, PSTConv2b, PSTConv3a, PSTConv3b, PSTConv4, and a fully-connected (FC) layer. In PSTConv1, PSTConv2a, PSTConv3a and PSTConv4, the spatial subsampling rate is set to 2 to halve the spatial resolution. In PSTConv2a and PSTConv3a, the temporal stride is set to 2 to halve the temporal resolution. The $r_o$ denotes the initial spatial search radius. The temporal padding $p$ is split as $[p_1, p_2]$, where $p_1$ and $p_2$ denote the left padding and the right padding, respectively. After these convolution layers, average and max poolings are used for spatial pooling and temporal pooling, respectively. The FC layer is then used as a classifier.

## B.2 PSTNet for 4D Semantic Segmentation

Figure 8: Hierarchical PSTNet architecture for 4D semantic segmentation.

As shown in Fig. 8, we employ 4 PST convolutional layers, *i.e.*, PSTConv1, PSTConv2, PSTConv3, and PSTConv4, and 4 PST transposed convolutional layers, *i.e.*, PSTConvTrans1, PSTConvTrans2, PSTConvTrans3, and PSTConvTrans4 for 4D Semantic Segmentation. The spatial subsampling rate is set to 4 to reduce the spatial resolution in each PST convolution layer. Following (Choy et al., 2019; Liu et al., 2019e), we conduct 4D semantic segmentation on clips with the length of 3 frames, *i.e.*, $L$ is fixed to 3. In PSTConv3 and PSTConvTrans2, the temporal kernel size is set to 3. Skip connections are added from input to PSTConvTrans1, PSTConv1 to PSTConvTrans2, PSTConv2 to PSTConvTrans3, and PSTConv3 to PSTConvTrans4. After PSTConvTrans4, a 1D convolution layer is appended for semantic predictions.

## C Implementation Details

To build a unified network and train the network with mini-batch, the number of spatial neighbors needs to be fixed. We follow existing point based works (Qi et al., 2017b; Liu et al., 2019e) to randomly sample a fixed number of neighbors. For 3D action recognition, the number is set to 9. For 4D semantic segmentation, we follow (Liu et al., 2019e) to set the number to 32. If the actual number of neighbors of a point is less than the set one (*e.g.*, 9), we randomly repeat some neighbors.

We train our models for 35 epochs with the SGD optimizer. Learning rate is set to 0.01, and decays with a rate of 0.1 at the 10th epoch and the 20th epoch, respectively.

MSR-Action3D (Li et al., 2010): Following (Liu et al., 2019e), batch size is set to 16 and frame sampling stride are set to and 1, respectively. We set the initial spatial search radius $r_o$ to 0.5.

NTU RGB+D 60 /120 (Shahroudy et al., 2016; Liu et al., 2019a): Following (Wang et al., 2020), batch size is set to 32. We set the clip length, frame sampling stride and the initial spatial search radius $r_o$ to 23, 2 and 0.1, respectively.

Synthia 4D (Choy et al., 2019): Following (Liu et al., 2019e), batch size and frame sampling stride are set to 12 and 1, respectively. We set the spatial search radius $r_o$ to 0.9.

We have implemented our PSTNets using both PyTorch and PaddlePaddle, which have shown similar performance.

## D Influence of Temporal Kernel Size on Different Actions

We show the influence of temporal kernel size on different actions in MSR-Action3D in Fig. 9. Clip length is 24.

When temporal kernel size $l = 1$, temporal correlation is not exploited. However, PSTNet can leverage pose information for recognition, especially for actions whose poses are unique in the dataset. For example, PSTNet with $l = 1$ can correctly recognize the "jogging" action because the action's pose is discriminative in the MSR-Action3D dataset. However, PSTNet with motion modeling ($l \geq 3$) can distinguish similar actions such as "draw x", "draw tick" and "draw circle". The results support our intuition that PSTNet effectively captures dynamics in point cloud sequences.

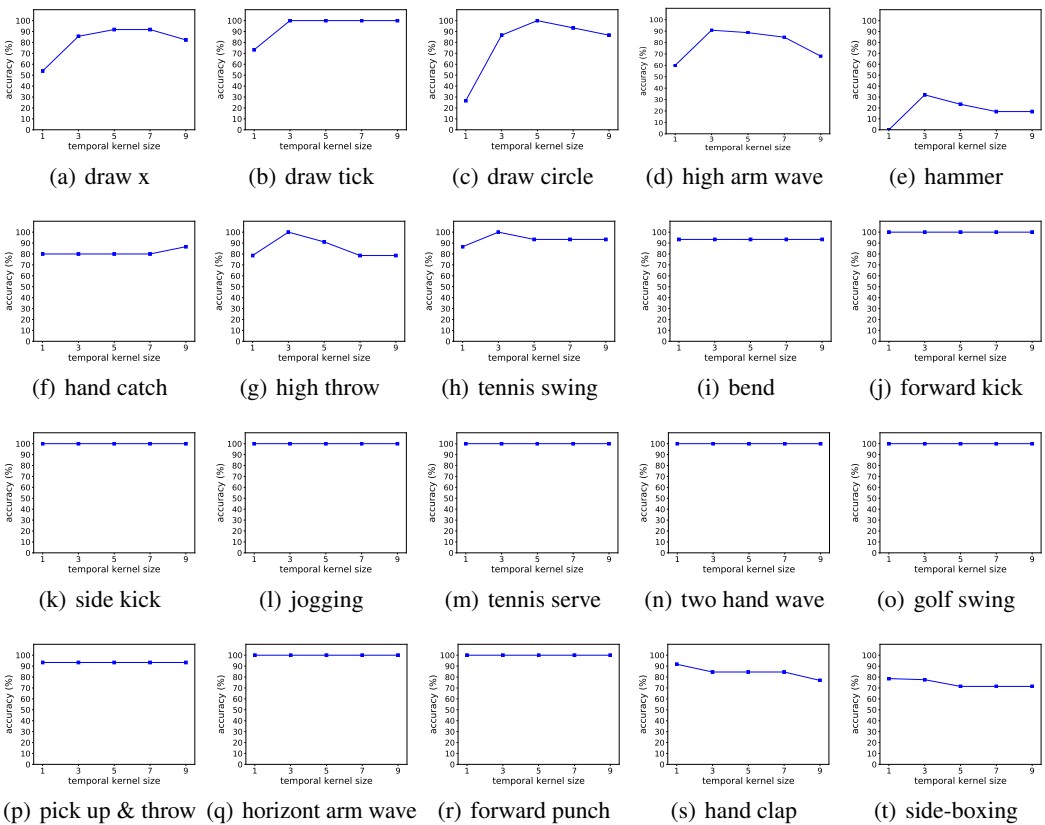

(a) draw x    (b) draw tick    (c) draw circle    (d) high arm wave    (e) hammer

(f) hand catch    (g) high throw    (h) tennis swing    (i) bend    (j) forward kick

(k) side kick    (l) jogging    (m) tennis serve    (n) two hand wave    (o) golf swing

(p) pick up & throw    (q) horizont arm wave    (r) forward punch    (s) hand clap    (t) side-boxing

Figure 9: Influence of temporal kernel size on different actions of MSR-Action3D.

# E  FEATURE VISUALIZATION

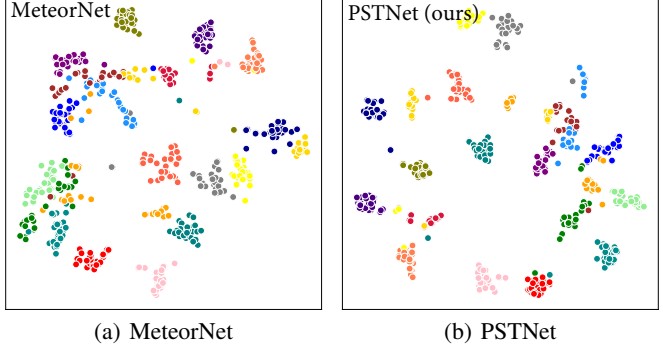

(a) MeteorNet      (b) PSTNet

Figure 10: Feature visualizations on MSR-Action3D using t-SNE. Each sequence is visualized as a point and sequences belonging to the same action have the same color. PSTNet features are semantically separable compared to MeteorNet, suggesting that it learns better representations for point cloud sequences.

We qualitatively evaluate PSTNet's ability to encode point cloud sequence by visualizing the learned features. We compare PSTNet with the sate-of-the-art MeteorNet (Liu et al., 2019e). Features are extracted from the layer before classifier (FC). These features are then projected to 2-dimensional space using t-SNE. As shown in Fig. 10, PSTNet feature is more compact and discriminative than MeteorNet.

# F  SYNTHIA 4D SEMANTIC SEGMENTATION RESULT DETAILS

We list the segmentation result for each class in Table 4. The Synthia 4D dataset contains 12 categories. Our PSTNet achieves five best accuracies among them, demonstrating the effectiveness of PSTNet.

Table 4: Semantic segmentation result (mIoU %) details on the Synthia 4D dataset.

| Method | # Frames | Bldn | Road | Sdwlk | Fence | Vegittn | Pole | Car | T. Sign | Pedstrn | Bicycl | Lane | T. Light | Average |
|---|---|---|---|---|---|---|---|---|---|---|---|---|---|---|
| 3D MinkNet14 (Choy et al., 2019) | 1 | 89.39 | 97.68 | 69.43 | 86.52 | 98.11 | 97.26 | 93.50 | 79.45 | 92.27 | 0.00 | 44.61 | 66.69 | 76.24 |
| 4D MinkNet14 (Choy et al., 2019) | 3 | 90.13 | 98.26 | 73.47 | 87.19 | **99.10** | 97.50 | 94.01 | 79.04 | **92.62** | 0.00 | 50.01 | 68.14 | 77.46 |
| PointNet++ (Qi et al., 2017b) | 1 | 96.88 | 97.72 | 86.20 | 92.75 | 97.12 | 97.09 | 90.85 | 66.87 | 78.64 | 0.00 | 72.93 | 75.17 | 79.35 |
| MeteorNet (Liu et al., 2019e) | 3 | **98.10** | 97.72 | 88.65 | 94.00 | 97.98 | **97.65** | 93.83 | **84.07** | 80.90 | 0.00 | 71.14 | **77.60** | 81.80 |
| PSTNet ($l=1$) | 3 | 96.32 | 98.07 | 85.40 | 94.66 | 97.16 | 97.51 | 94.83 | 76.65 | 76.99 | 0.00 | 75.39 | 76.45 | 80.79 |
| PSTNet ($l=3$) | 3 | 96.91 | **98.33** | **90.83** | **95.00** | 96.96 | 97.61 | **95.15** | 77.45 | 85.68 | 0.00 | **75.71** | 77.28 | **82.24** |

# G  PST CONVOLUTION WITHOUT DISENTANGLING SPACE AND TIME

To confirm the effectiveness of our disentangling structure for spatio-temporal modeling, we perform 3D action recognition on MSR-Action3D with non-decomposing convolution. To combine space and time, 3D displacements and time differences are encoded together to generate convolution kernel. We evaluate our PSTNet with different clip lengths. As shown in Fig. 11, the disentangling PST convolution achieves better accuracy than the non-decomposing structure, demonstrating the effectiveness of disentangling space and time for point cloud sequence modeling.

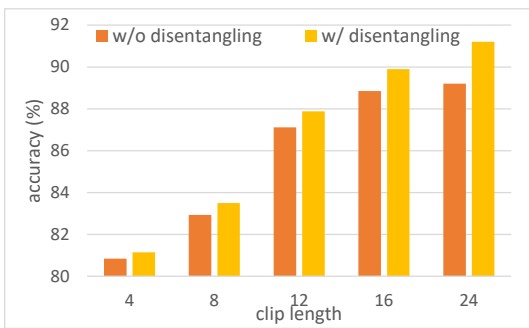

Figure 11: Comparison of PST convolution with (w/) and without (w/o) disentangling structure on MSR-Action3D.

# H  IMPACT OF DISPLACEMENT KERNEL AND SHARING KERNEL

In this paper, we design a lightweight implementation for the $f$ function in Eq. (5), *i.e.*, $f\big((\delta_x, \delta_y, \delta_z); \boldsymbol{\theta}\big) = \boldsymbol{\theta}_d \cdot (\delta_x, \delta_y, \delta_z)^T \cdot \mathbf{1} \odot \boldsymbol{\theta}_s$, where $\boldsymbol{\theta}_d$ is a displacement transform kernel, $\boldsymbol{\theta}_s$ is a sharing kernel. In this section, we investigate the influence of these two kernels. We conduct 3D action recognition on MSR-Action3D. The experimental results are shown in Table 5.

Table 5: Impact of displacement kernel and sharing kernel on 3D action recognition accuracy (%). The MSR-Action3D dataset is used.

| $\boldsymbol{\theta}_d$ | $\boldsymbol{\theta}_s$ | 16-frame | 24-frame |
|---|---|---|---|
| ✓ | | 58.25 | 61.67 |
| | ✓ | 86.35 | 87.46 |
| ✓ | ✓ | **89.90** | **91.20** |

Without $\boldsymbol{\theta}_s$, the spatial convolution degenerates into $\boldsymbol{M}_t^{(x,y,z)} = \sum_{\|(\delta_x,\delta_y,\delta_z)\| \leq r} \boldsymbol{\theta}_d \cdot (\delta_x, \delta_y, \delta_z)^T$. In this way, only the points in the final layer of our PSTNet are used. Therefore, the accuracy decreases dramatically.

Without $\boldsymbol{\theta}_d$, the spatial convolution becomes $\boldsymbol{M}_t^{(x,y,z)} = \sum_{\|(\delta_x,\delta_y,\delta_z)\| \leq r} \boldsymbol{\theta}_s \cdot \boldsymbol{F}_t^{(x+\delta_x,y+\delta_y,z+\delta_z)}$. In this fashion, the points in a neighborhood are treated equally. Point features are leveraged but their positions are ignored. Because spatial structure is not well captured, the accuracy decreases compared to the case where both $\boldsymbol{\theta}_d$ and $\boldsymbol{\theta}_s$ are used.

# I COMPUTATIONAL EFFICIENCY AND MEMORY USAGE

In this section, we evaluate the computational efficiency and memory usage, *i.e.*, the running time and number of parameters, of our method. We conduct 3D action recognition with a clip length of 16 on MSR-Action3D using 1 Nvidia Quadro RTX 6000 GPU.

As shown in Table 6, the proposed PSTNet is 41.5% faster than MeteorNet (Liu et al., 2019e). This is because that MeteorNet is based on MLPs, which are less efficient than convolutions.

Table 6: Comparison of parameter and running time on 3D action recognition accuracy (%). The MSR-Action3D dataset is used. Clip length is 16.

| Method | # Parameters (M) | Running time per clip (ms) | Accuracy (%) |
|---|---|---|---|
| MeteorNet (Liu et al., 2019e) | 17.60 | 54.56 | 88.21 |
| PSTNet (lightweight) | **8.44** | **31.92** | 89.90 |
| PSTNet (with PointConv) | 20.46 | 102.72 | **90.24** |

Second, we study the impact of the number of PSTNet layers on running time and parameter. As shown in Fig. 12, when PSTNet becomes deep, the number of parameters significantly grows. This is because, like most deep neural networks, the point feature dimension exponentially increases to improve the feature representation ability in PSTNet, which needs more parameters. However, the running time does not increase significantly. This is because our PSTNet is spatio-temporally hierarchical, points are exponentially reduced along both spatial and temporal dimensions, thus saving running time.

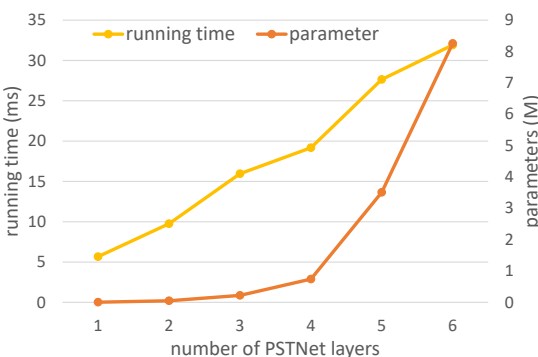
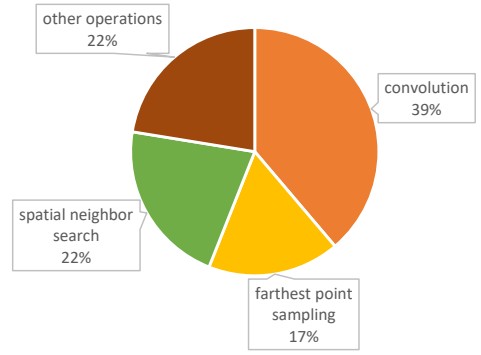

Figure 12: Impact of the number of PSTNet layers on running time and parameter.

Figure 13: Running time proportion of main operations in PSTNet.

Third, we investigate the running time of main operations in PSTNet, including convolution, farthest point sampling and spatial neighbor search. We show the running time proportion of these operations in Fig. 13.

Finally, because the $f$ function in Eq. (5) can be implemented in different ways, we replace our lightweight spatial convolution with PointConv (Wu et al., 2019) to investigate the influence of the spatial modeling in our PSTNet. As shown in Table 6, using PointConv only improves accraucy slightly compared to our lightweight spatial convolutions, but significantly increases parameters and

running time. This is because PointConv utilizes MLP based operations to process points and their features, which are less efficient than our lightweight convolutions.

## J   IRREGULAR FRAME SAMPLING

Our PSTNet can also process irregularly sampled point cloud sequences by frame interpolation. In this section, we randomly remove some frames in sequences. Specifically, we conduct 3D action recognition on MSR-Action with 24 frames. We randomly remove 8 frames from each clip. Then, we use replication and Iterative Closest Point (ICP) (Besl & McKay, 1992) to interpolate missing frames, respectively. We compare our PSTNet with MeteorNet (Liu et al., 2019e), which explicitly encodes time and does not need interpolation.

Table 7: 3D action recognition on MSR-Action3D with irregularly sampled frames. Clip length is originally 24 and then 8 frames are randomly removed from each clip.

| Method | Accuracy (%) |
|---|---|
| MeteorNet (Liu et al., 2019e) | 88.40 |
| PSTNet (replication interpolation) | 90.24 |
| PSTNet (ICP interpolation) | **90.94** |

As shown in Table 7, our PSTNet still achieves the best accuracy, indicating that PSTNet is able to model point cloud sequences with irregular frame sampling.

## K   SCENE FLOW ESTIMATION

To evaluate our method on the LiDAR data where points in a frame can be highly irregular, we apply our method to scene flow estimation. We follow the setting proposed by MeteorNet (Liu et al., 2019e), *i.e.*, given a point cloud sequence, estimating a flow vector for every point in the last frame. However, because our point tube is constructed according to the middle frame, which is not applicable to the last-frame scene flow estimation, we adapt temporal anchor frame selection and spatial anchor point transferring. Specifically, we select the last frame in each tube as the anchor frame. Then, after spatial sampling, each anchor point is transferred to its previous nearest $l$ frames. Following MeteorNet, we first train our model on a FlyingThings3D dataset according to the synthetic method in (Liu et al., 2019e), and then fine-tune the model on a KITTI scene flow dataset (Liu et al., 2019e). Point tracking is not used.

Table 8: Scene flow estimation accuracy on the KITTI scene flow dataset.

| Method | Input | # Frames | End-Point-Error |
|---|---|---|---|
| FlowNet3D (Liu et al., 2019d) | points | 2 | 0.287 |
| MeteorNet (Liu et al., 2019e) | points | 3 | 0.282 |
| PSTNet (ours) | points | 3 | **0.278** |

As shown in Table 8, our PSTNet achieves promising accuracy on scene flow estimation.

## L   MOTION RECOGNITION ON THE SYNTHETIC MOVING MNIST POINT CLOUD DATASET

In the real world, it is impossible to obtain point IDs. To evaluate our PSTNet in the scenario where points are correlated and can be accurately tracked across frames, we conduct motion recognition on a synthetic Moving MNIST Point Cloud dataset. To synthesize moving MNIST digit sequences, we use a generation process similar to that described in (Fan & Yang, 2019). Each synthetic sequence consists of 16 consecutive point cloud frames. Each frame contains one handwritten digit moving and bouncing inside a $64 \times 64$ area. The digits are chosen randomly from the MNIST dataset. We

use the same training/test split as the original MNIST dataset. For each digit, we sample 128 points. Point order is maintained across frames so that tracking can be employed. We design 144 motions, including

- nine initial locations: $(1, 1), (1, 3), (1, 5), (3, 1), (3, 3), (3, 5), (5, 1), (5, 3), (5, 5)$.
- eight velocities: $(1, 1), (-1, 1), (-1, -1), (1, -1), (2, 2), (-2, 2), (-2, -2), (2, -2)$.
- two kinds of digit distortion: scaling digits horizontally or vertically, *i.e.*, $(|0.4 - 0.05t| + 0.6) \times (x - c_x) + c_x$ or $(|0.4 - 0.05t| + 0.6) \times (y - c_y) + c_y$, where $(c_x, c_y)$ is the digit center and $t \in [1, 16]$.

A few motion examples are shown in Fig. 14. Because motion and appearance are independent of each other, it is challenging to recognize motion while avoiding interference from digit appearance.

Figure 14: Motion examples in the synthetic Moving MNIST Point Cloud dataset.

To exploit point correlation, anchor points and their neighbors are selected in the first frame, which are then propagated to other frames. In this fashion, anchors and their neighbors are tracked across frames.

Table 9: Motion recognition accuracy on the Moving MNIST Point Cloud dataset.

| Method | Accuracy (%) |
|---|---|
| PSTNet (original) | 97.15 |
| PSTNet (with tracking) | **98.74** |

As shown in Table 9, both the original PSTNet and the tracking based PSTNet achieve promising accuracy on the Moving MNIST Point Cloud dataset. Our original PSTNet achieves similar accuracy with the tracking based PSTNet in this simulated case, demonstrating that our PSTNet does not heavily rely on point IDs or tracking.

# M  VISUALIZATION OF THE OUTPUT OF EACH PST CONVOLUTION LAYER IN PSTNET

We visualize the output of each layer in the PSTNet architecture trained on MSR-Action3D in Fig. 15. Because we use ReLU as the activation function, all outputs are greater than zero and large outputs indicate high activation. To visualize outputs, we squeeze output vectors to scalars via $l_1$ norm. We observe that:

- For PSTConv1, because temporal kernel size $l = 1$, it does not capture the temporal correlation. In this case, PSTConv1 focuses on the appearance and therefore outputs high activation to performer contours.
- PSTConv2a to PSTConv4. When modeling the spatio-temporal structure of point cloud sequence, PST convolutions mainly focus on moving areas so as to achieve informative representations of actions.

The visualization results support our intuition that the proposed PST convolution effectively captures dynamics of point cloud sequences.

## N  VISUALIZATION OF 4D SEMANTIC SEGMENTATION

We visualize segmentation results from the Synthia 4D dataset in Fig. 16 and Fig. 17. Our PSTNet can accurately segment most objects.

## O  LIMITATION

Like most deep neural networks, the proposed PSTNet relies on large-scale labeled datasets for training. A potential improvement is to integrate some learning methods, *e.g.*, few-shot learning (Liu et al., 2019c;f; Zhu & Yang, 2020; Liu et al., 2021), into point cloud sequences modeling to reduce the reliance on human-annotated data.

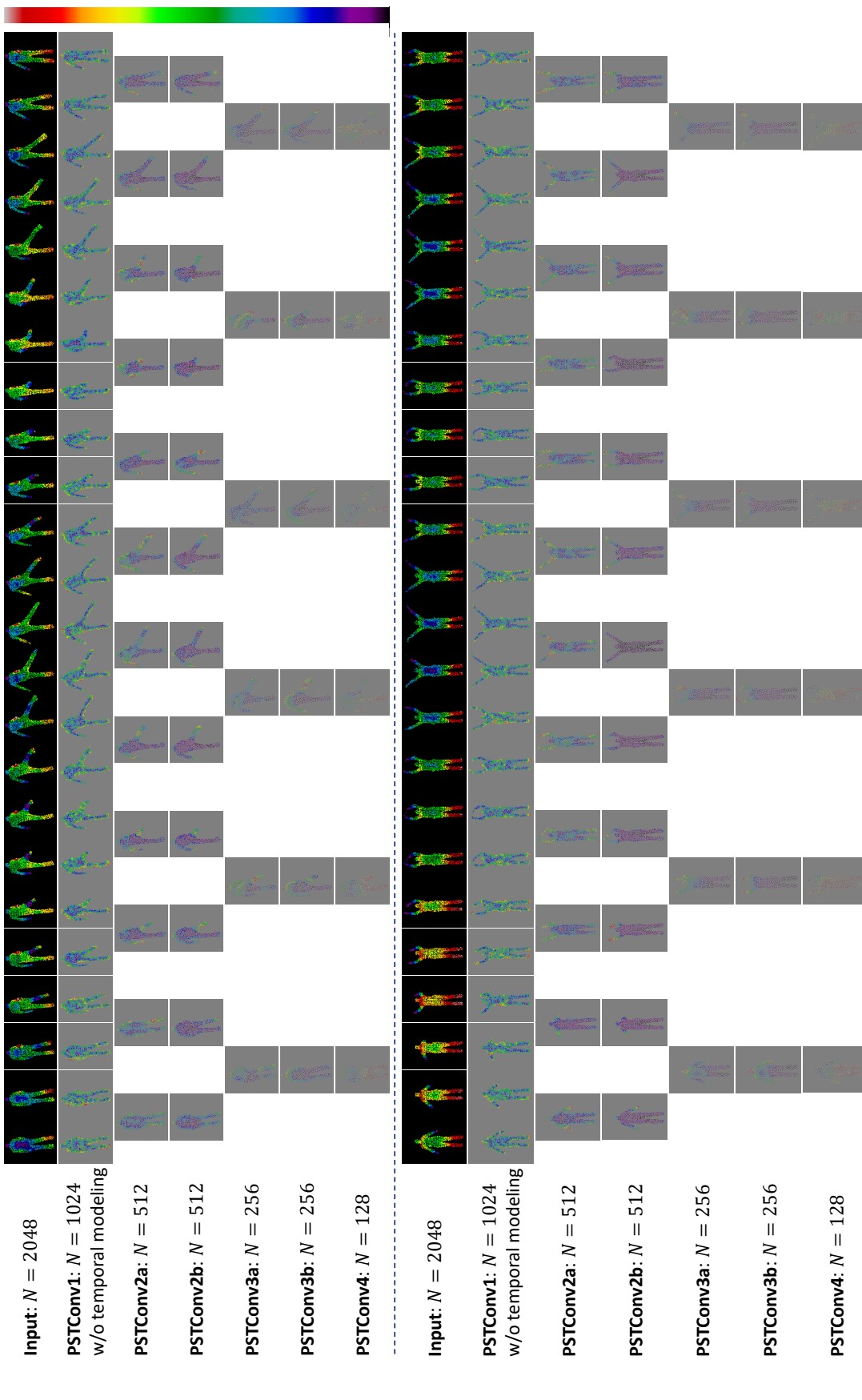

Figure 15: Visualization of the output of each layer in PSTNet. For the input point cloud sequence, color encodes depth. For the outputs, brighter color indicates higher activation. Input sequences consist of 24 frames. Due to the spatial subsampling $s_s$ and the temporal stride $s_t$, points and frames progressively decrease along the network depth.

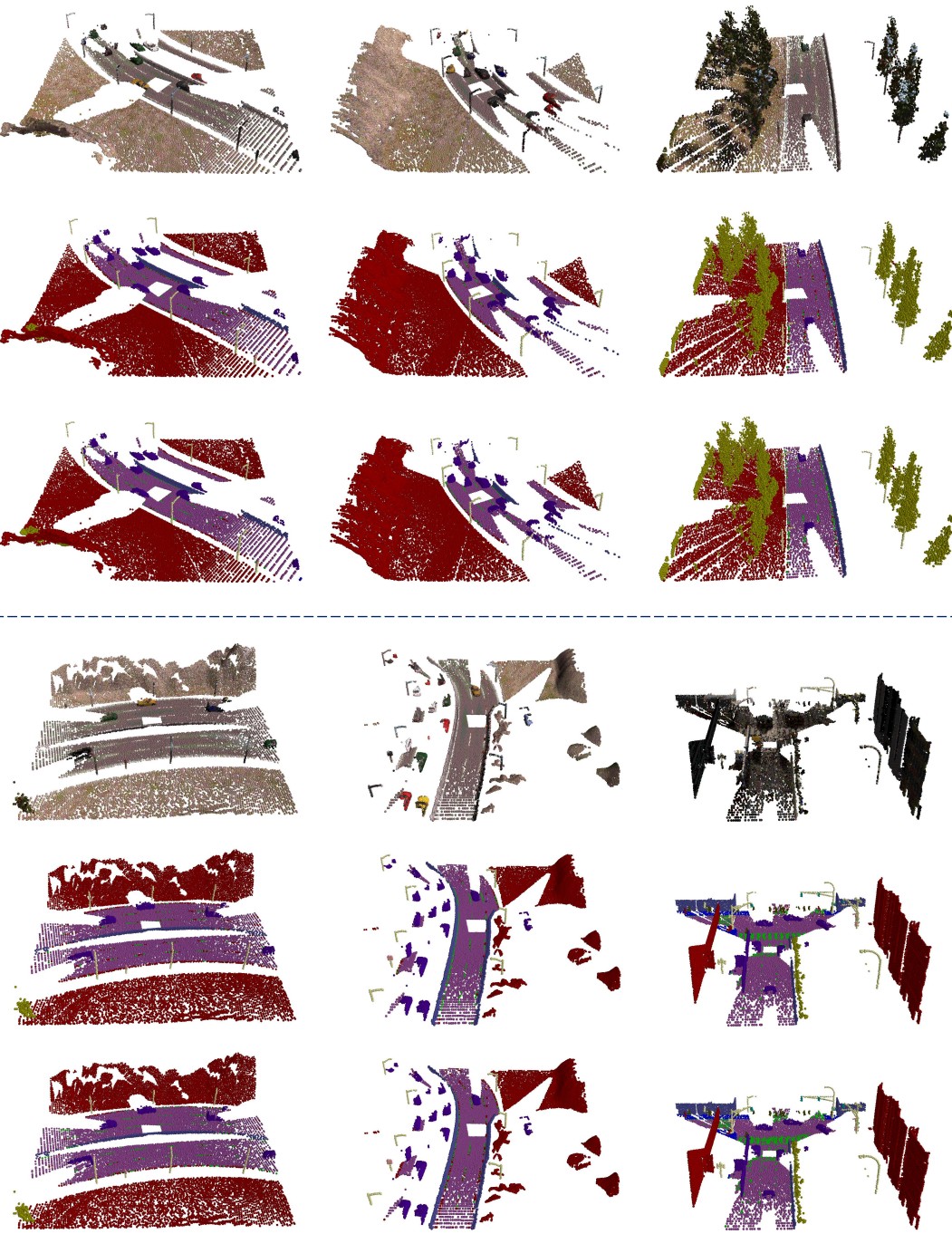

Figure 16: Visualization of semantic segmentation examples from Synthia 4D (I). Top: inputs. Middle: ground truth. Bottom: PSTNet predictions.

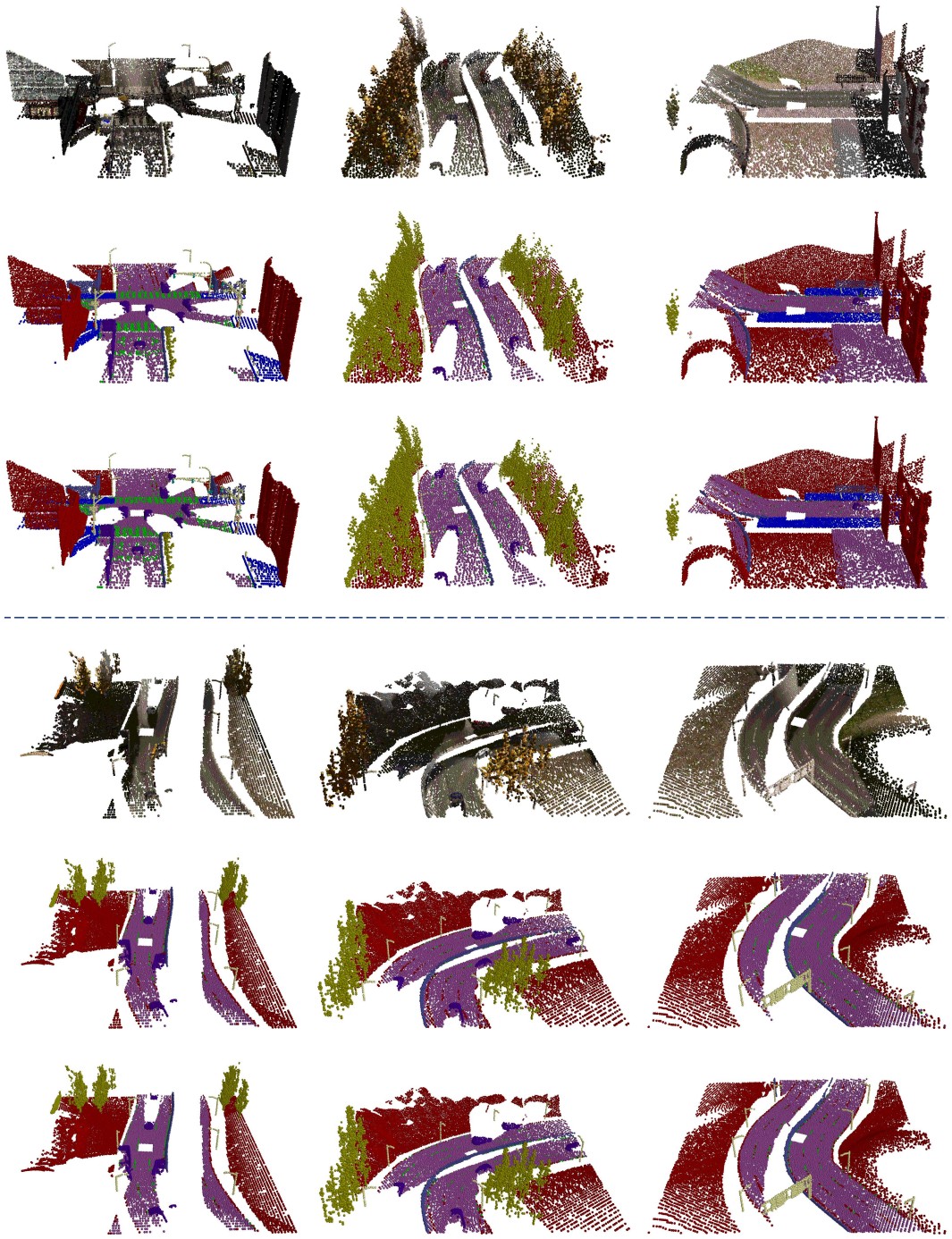

Figure 17: Visualization of semantic segmentation examples from Synthia 4D (II). Top: inputs. Middle: ground truth. Bottom: PSTNet predictions.

