# OpenReview forum: "PSTNet: Point Spatio-Temporal Convolution on Point Cloud Sequences"
_ICLR.cc/2021/Conference — ICLR 2021 Poster_

### Official Review · AnonReviewer2 · 2020-10-25
**Promising results and well elaborated**

**Rating:** 7
**Confidence:** 3

**Review:**

The paper introduces point spatio-temporal convolutions, which are used for the feature extraction of point cloud sequences. A trainable kernel is used which is applied locally as a continuous convolution. An important aspect is that the temporal dimension is processed separately, with an additional convolution, instead of simply using a 4D convolution. The authors claim that in this way the network will achieve a better understanding of the dynamics of the input.

The presented method is very interesting in my eyes. I find the argumentation concerning the construction of the spatio-temporal conclusive and it is supported by the experiments. Together with the presented striding, based on farthest point sampling, and the transposed convolution, the method seems to be very versatile, as it was shown in the results. However, a few points are still not quite clear to me:
* Why is striding used for the semantic segmentation test? Theoretically, this problem could be solved much easier without striding and transposed convolutions.
* The results were mainly sequences where the points of the frames do not correlate. What about data where the points correlate, like in physical simulations? Would the presented point tubes make sense there?

The paper is written understandably and the evaluations are sufficient, IMO, to confirm the claims given by the authors. Maybe the following papers could be relevant as related work: Occupancy Flow: 4D Reconstruction by Learning Particle Dynamics and Tranquil Clouds: Neural Networks for Learning Temporally Coherent Features in Point Clouds, since they also deal with point sequences.
In section 3.1 there is a small typo (... CNNs usually adopt small ~~kennel~~ **kernel** sizes ...), just like at the beginning of section 4.3 (Temporal ~~kennel~~ **kernel** size...).

All in all I find the method simple but promising. The paper seems well elaborated and I would tend to accept it.

---

> ### Author Response · Authors · 2020-11-21
> **Reason for temporal modeling in the semantic segmentation and correlated points**
>
> We thank you for acknowledging that our method is  **“very interesting”** and **“very versatile”**, and our results are **“promising”**. We also thank you for your valuable comments.
>
> **1.  Reason for temporal modeling in the semantic segmentation**
>
> When capturing temporal correlation, our PSTNet involves a temporal kernel size for temporal correlation modeling and a temporal stride for frame subsampling. We follow MinkNet14 and MeteorNet, and take 3 frames as input. Since the clip is too short, we do not exploit striding to subsample frames, but set temporal kernel size to 3 to capture temporal correlation. Although semantic segmentation can be achieved from a single frame without temporal correlation, by exploiting temporal information we can constrain the consistency of segmented scene structure across frames, thus improving segmentation accuracy and robustness to noise.
>
> **2.  Correlated points**
>
> When points are correlated across frames, both anchors and neighbors can be accurately tracked. In this case, our point tubes can be constructed according to tracked anchor points.
>
> In the real world, it is impossible to obtain point IDs, because points emerge inconsistently and may flow in and out. To evaluate our PSTNet in the scenario where points can be tracked across frames, we conduct **motion recognition**  on a synthetic Moving MNIST Point Cloud dataset. Each synthetic sequence consists of 16 consecutive point cloud frames.  Each frame contains one handwritten digit moving and bouncing inside a $64 \times 64$ area. The digits are chosen randomly from the MNIST dataset. We sample 128 points for each digit. The point order is maintained across frames so that tracking can be employed.  We design 144 motions, including 9 initial locations, 8 velocities and 2 kinds of digit distortion.  Because motion and appearance are independent of each other, it is challenging to recognize motion while avoiding interference from digit appearance.  We use the same training/test split as the original MNIST dataset. For details, please refer to Appendix Section P.
>
> To exploit point correlation, anchor points and their neighbors are selected in the first frame, which are then propagated to other frames.  In this fashion, anchors and their neighbors are tracked across frames.
>
> $\ \ \ \ \ $Table 1  (Appendix Section P)
>
> | Method$\ \ \ \ \ \ \ \ \ \ \ \ \ \ \ \ \ \ \ \ \ \ \ $ | Acc (%)  |
>
> | PSTNet (original)  $ \ \ \ \ \ \ \ \ $ | $\ \ $97.15$\ $ |
>
> | PSTNet (with tracking)  | $\ \ $98.74$\ $ |
>
> As shown in Tab. 1,  both original and tracking based PSTNet achieve promising accuracy on the Moving MNIST Point Cloud dataset. Our original PSTNet achieves similar accuracy with tracking based PSTNet in this simulated case, demonstrating that our PSTNet does not heavily rely on point IDs.
>
> **3.  Related work**
>
> Thank you for pointing out these two interesting works that also attempt to model spatio-temporal point clouds. We have discussed them in our revision. Niemeyer [1] learned a temporal and spatial vector field in which every point is assigned with a motion vector of space and time for 4D reconstruction. Prantl et al. [2] learned stable and temporal coherent feature spaces for point-based super-resolution.
>
> [1] Michael Niemeyer, Lars M. Mescheder, Michael Oechsle and Andreas Geiger. Occupancy Flow: 4D Reconstruction by Learning Particle Dynamics. ICCV (2019)
>
> [2] Lukas Prantl, Nuttapong Chentanez, Stefan Jeschke and Nils Thuerey. Tranquil Clouds: Neural Networks for Learning Temporally Coherent Features in Point Clouds. ICLR (2020)

---

### Official Review · AnonReviewer3 · 2020-10-30
**PSTNet: Point Spatio-Temporal Convolution on Point Cloud Sequences**

**Rating:** 5
**Confidence:** 5

**Review:**

This paper aims to process the point cloud data in a convolution manner. The authors propose the PST convolution and deconvolution operations to handle the different tasks such as the classification and segmentation on point cloud. The extensive experiments verify the effectiveness of the proposed method and achieve state-of-the-art results on multiple benchmark datasets. Overall, the paper is well-written and organized.

Strength:
 •	In this paper, the authors introduce a method named PST convolution which could directly handle the sequential point cloud data. The PST convolution could capture the spatio-temporal in a convolution manner. They decompose the spatial and temporal information and conduct convolution on them respectively.
•	In order to handle both the abstraction level and point-wise level tasks, they build a complete convolution system including convolution and deconvolution. Also, the point tube concept is introduced here to preserve the spatio-temporal structure.
•	The experiments are exhaustive and impressive. Especially the author adopts various kinds of demonstration to show the effectiveness.

 Weakness:
 •	Even through the experiments results look pretty good, the novelty of this paper is still limited. In this paper, compared to the previous paper such as MeteorNet, the authors explicitly promotes the concept of capturing the spatial and temporal information. There is no significant difference with MeteorNet or even PointNet++. The MeteorNet also extracts the spatio-temporal information by chain-flow radius search. No matter the point tube or the anchor points extracted in the spatial and temporal domain, the main idea is finding a good neighbor for the current point. It should be described clearly in details for the novelty in the future revised version.
•	Also, even the convolution adopted in this paper could make the point processing more convenient, I have a concern about the efficiency of the proposed method. The computation cost including stacked convolution layers, the FPS sampling, and the nearest neighbor search are all heavy. So the authors may provide some efficiency comparison in the future.

---

> ### Author Response · Authors · 2020-11-21
> **Highlight differences between PSTNet and prior works and efficiency analysis (Part 1/2)**
>
> We thank you for acknowledging **“the extensive experiments verify the effectiveness”**  of our method and **“achieve state-of-the-art results on multiple benchmark datasets.”** We also thank you for the suggestion of efficiency comparison.
>
> **1. Comparison to MeteorNet or PointNet++**
>
> To clarify the differences, we formulate PointNet++, MeteorNet and our method as follows,
>
> PointNet++:         $\textit{F}'^{(x,y,z)} = \mathrm{MAX}\_{||(\delta_{x}, \delta_y, \delta_z)|| \le r}   \mathrm{MLP}(\textit{F}^{(x,y,z)}, \textit{F}^{(x+\delta_x,y+\delta_y,z+\delta_z)}, \delta_{x}, \delta_y, \delta_z)$
>
> MeteorNet:          $\textit{F}’^{(x,y,z)}\_t = \mathrm{MAX}\_{||(\delta_{x}, \delta_y, \delta_z)|| \le r}\mathrm{MLP}(\textit{F}_t^{(x,y,z)}, \textit{F}_\{t+\delta\_t}^{(x+\delta_x,y+\delta_y,z+\delta_z)}, \delta_x,\delta_y,\delta_z, \delta_t)$
>
> PSTNet (ours):     spatial         $\ \ \ \ \textit{M}^{(x,y,z)}\_t = \sum\_{||(\delta_{x}, \delta_y, \delta_z)|| \le r} \textit{S}^{(\delta_{x}, \delta_{y}, \delta_{z})}  \cdot \textit{F}^{(x+\delta_x,y+\delta_y,z+\delta_z)}\_t$
>
> $\ \ \ \ \ \ \ \ \ \ \ \ \ \ \ \ \ \ \ \ \ \ \ $temporal    $\textit{F}’^{(x,y,z)}\_t = \sum\_{k=-\lfloor l/2 \rfloor}^{\lfloor l/2 \rfloor}    \textit{T}\_k \cdot \textit{M}^{(x,y,z)}\_{t+k}$
>
> where $r$ is spatial search radius and $l$ is temporal kernel size.  Note that, MeteorNet does not involve temporal search radius or temporal kernel size.
>
> First of all, PointNet++ only models spatial information of static point clouds and does not have a mechanism to model temporal information of point cloud sequences. On the contrary, our PSTNet and MeteorNet are designed to address dynamic point clouds.
>
> Furthermore, there are three major differences between our proposed PSTNet and MeteorNet:
>
> - **Decoupling spatio-temporal information for point cloud sequence modeling.**
>   - Considering space and time are orthogonal and independent of each other, our PSTNet decouples spatio-temporal information of point cloud sequences. This decomposition significantly facilitates us to model the spatial and temporal information explicitly according to their distinct characteristics. Specifically, our PSTNet models temporal dynamics by leveraging the temporal order while addressing the spatial structure of point clouds by taking their spatial irregularity into account. On the contrary, MeteorNet simply stacks timestamps and coordinates together, and neglects the temporal order in timestamps. Thus, it may not well exploit temporal information, leading to inferior performance.
>   - The scales of spatial displacements and temporal differences in point cloud sequences may not be compatible. Treating them equally like MeteorNet is not in favor of network optimization and thus leads to inferior feature extraction. In contrast, by modeling these two modalities separately, our PSTNet achieves superior feature representation ability.
> - **Hierarchical spatio-temporal modeling benefiting from our point tube.**
> The grouping in MeteorNet only considers spatial radius. As MeteorNet neglects the local dependencies of neighboring frames, it models temporal information by using the sequence length as its temporal receptive field.
> Without constructing temporal hierarchy with temporal kernels, pooling or stride, MeteorNet faces two issues.
> (i) As points flow in and out of the region, especially for long sequences and fast motions, embedding points in a spatially local area along an entire sequence handicaps capturing instant local dynamics of point clouds in MeteorNet. (ii) Without the temporal stride or pooling, MeteorNet needs to process all the frames in each layer and thus is not efficient. In contrast, benefiting from our point tube, the proposed PSTNet is both spatially and temporally hierarchical, and thus is efficient to model long sequences. As shown in Tab. 1, from 16 frames to 24 frames, our method achieves 1.30% improvement while MeteorNet only obtains 0.29% improvement, demonstrating our method is more effective in modeling longer sequences.
>
> $\ \ \ \ \ \ \ \ \ \ \ \ \ \ \ \ \ \ \ \ \ \ \ \ \ \ \ \ \ \ \ \ \ \ \ \ $Table 1 (Appendix Section M)
>
> $\ \ \ \ \ \ $| Method$\ \ \ \ \ \ $|  # Params | 16-frame time | 16-frame acc | 24-frame acc |
>
> $\ \ \ \ \ \ $| MeteorNet | $\ \ $17.60 M$\ $ | $\ \ \ \ \ $54.56 ms$\ \ \ $ | $\ \ \ \ \ $88.21%$\ \ \ \ \ $ | $\ \ \ \ \ $88.50%$\ \ \ \ $ |
>
> $\ \ \ \ \ \ $| PSTNet$\ \ \  \ \ \ \$  | $\ \ \ $8.26 M $\ $   | $\ \ \ \ $31.92 ms $\ \ \ $ | $\ \ \ \ \ $89.90%$\ \ \ \ \ $ | $\ \ \ \ \ $91.20%     $\ \ \ $ |
>
> - **Operators.** To the best of our knowledge, our PSTNet is the first attempt to construct convolutions on raw point cloud sequences in a unified framework. Our convolution-based network outperforms MLP-based MeteorNet on all the tasks. Moreover, as shown in Tab. 1, our PSTNet uses fewer parameters and takes less time than MeteorNet, demonstrating our convolutional operations are effective and efficient.

---

> > ### Author Response · Authors · 2020-11-21
> > **Highlight differences between PSTNet and prior works and efficiency analysis (Part 2/2)**
> >
> > Overall, we believe that our PSTNet is significantly different from MeteorNet or PointNet++. Note that radius/neighboring search and FPS sampling are commonly-used operations in point cloud processing. In this paper, we did not claim they are our contribution. The decomposition of spatio-temporal information in representing dynamic point clouds and the hierarchical temporal modeling are our major contributions.
> >
> > **2.  Efficiency of stacking multiple layers**
> >
> > Stacking multiple layers to form hierarchical architectures is a common technique in deep neural networks, not only used in our PSTNet. In Fig. 14, the running time of PSTNet with different numbers of layers are shown as follows,
> >
> > $\ \ \ \ \ \ \ \ \ \ \ \ \ \ \ \ \ \ \ $Table 2 (Appendix Section M)
> >
> > | # layers $\ $ | $\ \ $1$\ \ $ | $\ \ \ $2$\ \ $ | $\ \ \ $3$\ \ \ $ |  $\ \ \ $4 $\ \ \ $ |  $\ \ $5 $\ \ \ $ |  $\ \ \ $6 $\ \ \ $ |
> >
> > |time (ms) |  5.67 | 9.76 | 15.95 | 19.17 | 27.64 | 31.92 |
> >
> > As PSTNet becomes deep, its running time does not increase dramatically. Because our PSTNet is spatio-temporally hierarchical, points are exponentially reduced along both spatial and temporal dimensions, thus saving running time.
> >
> > **3.  Efficiency of FPS sampling and nearest neighbor search**
> >
> > First, our spatial convolutions use radius search instead of nearest neighbor search. Compared with nearest neighbor search, radius search does not need time-consuming sort. The running time of main operations in our PSTNet is shown in Tab. 3.
> >
> > $\ \ \ \ \ \ \ \ \ \ $Table 3  (Appendix Section M)
> >
> > | operation $\ \ \ \ \ $ | time (ms) | percentage |
> >
> > | convolution $\ \ $ | $\ \ \ $12.45$\ \ \ \ $ | $\ \ \ \ \ \ $39%$\ \ \ \ \ $ |
> >
> > | FPS sampling | $\ \ \ \ \ $5.43$\ \ \ \ $ | $\ \ \ \ \ \ $17%$\ \ \ \ \ $ |
> >
> > | radius  search | $\ \ \ \ \ $7.02$\ \ \ \ $ | $\ \ \ \ \ \ $22%$\ \ \ \ \ $ |
> >
> > | others $\ \ \ \ \ \ \ \ \ \ $ |$\ \ \ \ \ \ $7.03$\ \ \ \ $ | $\ \ \ \ \ \ $22%$\ \ \ \ \ $ |
> >
> > The running time is mainly spent on convolutions.  Note that, if replacing our convolutions with MLPs, the MLP operations take longer time, i.e., 32.78 ms. Then, the running time percentage is MLP : FPS : radius search: others = 63% : 10% : 13.5% : 13.5%.  Therefore, in point cloud processing, the main running time is not caused by FPS sampling or radius search.

---

### Official Review · AnonReviewer1 · 2020-11-02
**Principled and clearly presented; empirical justification needs improvement.**

**Rating:** 7
**Confidence:** 4

**Review:**

This paper introduces a new convolutional approach to directly process raw spatiotemporal (ST) point cloud data. The proposed point spatio-temporal (PST) convolution operates on "point tubes" and decouples space and time through a shared spatial convolution at each timestep, followed by a temporal convolution. It also introduces a transposed PST to enable point-wise predictions in an encoder-decoder framework (PSTNet). The presented experiments demonstrate the effectiveness of these convolutions by using PSTNet for action recognition and semantic segmentation on point cloud sequences, showing improvement over relevant recent work.

Strengths:
+ The approach is technically novel. Processing raw 4D point clouds of arbitrary length is a relatively new direction, and I have not seen a convolutional approach before.
+ I like the idea of decoupling spatial and temporal convolutions. As the paper points out, this should allow better handling of varied spatial sampling over time.
+ Though the ST neighborhood size (i.e. temporal kernel size and spatial search radius) must be manually set and tuned, these neighborhoods are defined and structured in a principled way similar to grid-based CNNs which could make tuning easier than in prior work (e.g. MeteorNet).
+ Experiments show slight improvement over the most relevant SOTA MeteorNet and MinkowskiNet for action recognition and semantic segmentation from depth sequences. I also appreciate the analysis of sensitivity to ST neighborhood parameters and the informative visualization of learned features (Fig 4, 12, and 13).
+ The paper is, for the most part, well-written and nicely presented (e.g. Fig 2). The detailed supplementary material and promised code release enabled reproducibility.

Weaknesses:
- The paper could better situate the proposed method in the context of relevant prior work (see "Related Work" below).
- The proposed PST convolution is described as "generic", but as far as I can tell it would struggle with irregular temporal sampling. The architecture operates on ordered point cloud sequences but does not leverage timestamps, so there is an underlying assumption that point clouds are sampled at a fixed rate that does not change from training to test time.
- Various claims and design decisions could be better supported through additional experiments; in particular, the choice to decouple space and time and the spatial kernel design (see "Experiments" below).
- There is no discussion about the limitations of the proposed method in the paper.

Based on these points and the ones detailed below, I initially lean towards accepting the paper. The proposed method is technically novel and principled, and the experiments are sufficient to show PSTNet can extract useful features. The ability to handle irregular timestamps and additional experiments to characterize the advantages of space/time decoupling would improve the paper, but their absence does not outweigh the contributions for me. My other concerns about related work and a discussion of limitations can be addressed in a revision.

Related work:
- PSTNet should be compared and contrasted to MeteorNet in more detail since this is the most relevant prior work. Currently, MeteorNet is described in an oversimplified way as applying PointNet++ to 4D points, but there are actually many similarities to the proposed method ("point tubes" are analogous to neighborhood grouping and the PST convolution to PointNet aggregation).
- The proposed spatial kernel (Eq 5) is not compared to or motivated by related static point cloud convolutional methods in the Related Work or Methods sections. Additionally, some important relevant works are not cited, e.g., [1][2].
- Other relevant works that consider spatiotemporal point clouds in a less general setting may be worth mentioning: e.g. in scene flow estimation [3] or learning from spatiotemporal object point clouds [4].

[1] PointCNN: Convolution on X-Transformed Points, Li et al., 2018

[2] Dynamic Graph CNN for Learning on Point Clouds, Wang et al., 2019

[3] HPLFlowNet: Hierarchical Permutohedral Lattice FlowNet for Scene Flow Estimation on Large-scale Point Clouds, Gu et al., 2019

[4] CaSPR: Learning Canonical Spatiotemporal Point Cloud Representations, Rempe et al., 2020

Experiments:
- The decision to decouple spatial/temporal convolutions would be more convincing with: (1) a comparison to a baseline PST convolution that does not fully decompose space and time (Eq 2), and (2) an evaluation on LIDAR data where spatial sampling can be highly irregular and variable over time. For example, the task of scene flow estimation from LIDAR would indicate if PSTNet can deal with high spatial irregularity while still encoding necessary local details.
- The design of the learned spatial kernel f (Eq 5 and following paragraph) is not extensively motivated or justified in the text or by experimental results. How does the displacement/sharing kernel design affect performance or compare to using an MLP or other prior work like PointConv which considers point density?
- If possible, a runtime comparison to MeteorNet/MinkowskiNet would be useful since these are similar generic ST point processing approaches.

Minor Suggestions:
- The formulation in Sec 3.2 unclear and inconsistent at times: e.g. (1) the displacement vector is described as a real number following Eq 3, but as a vector following Eq 5, and (2) the shape of F is not detailed for Eq 2-5. In general, I think the exposition in Sec 3.2 would be improved by intuitively describing the "Point Tube" idea before the precise mathematical formulation.
- In Fig 5, GPU runtime is longer than CPU?
- Typo in Sec 4.3, second paragraph: "kennel size" -> "kernel size".

---

> ### Author Response · Authors · 2020-11-21
> **More discussions and experiments (Part 1/2)**
>
> We thank you for acknowledging our work is **“technically novel and principled”** and experiments are **“sufficient to show PSTNet can extract useful features”**. We are glad to see that you **“like the idea of decoupling spatial and temporal convolutions”**.  We also thank you for your constructive and detailed comments.
>
> **1. Differences from MeteorNet**
>
> There are three major differences between our proposed PSTNet and MeteorNet, as follows:
> - **Decoupling spatio-temporal information for point cloud sequence modeling.**
>   - Considering space and time are orthogonal and independent of each other, our PSTNet decouples spatio-temporal information of point cloud sequences. This decomposition significantly facilitates us to model the spatial and temporal information explicitly according to their distinct characteristics. Specifically, our PSTNet models temporal dynamics by leveraging the temporal order while addressing the spatial structure of point clouds by taking their spatial irregularity into account. On the contrary, MeteorNet simply stacks timestamps and coordinates together, and neglects the temporal order in timestamps. Thus, it may not well exploit temporal information, leading to inferior performance.
>   - The scales of spatial displacements and temporal differences in point cloud sequences may not be compatible. Treating them equally like MeteorNet is not in favor of network optimization and thus leads to inferior feature extraction. In contrast, by modeling these two modalities separately, our PSTNet achieves superior feature representation ability.
> - **Hierarchical spatio-temporal modeling benefiting from our point tube.**
> The grouping in MeteorNet only considers spatial radius. As MeteorNet neglects the local dependencies of neighboring frames, it models temporal information by using the sequence length as its temporal receptive field.
> Without constructing temporal hierarchy with temporal kernels, pooling or stride, MeteorNet faces two issues.
> (i) As points flow in and out of the region, especially for long sequences and fast motions, embedding points in a spatially local area along an entire sequence handicaps capturing instant local dynamics of point clouds in MeteorNet. (ii) Without the temporal stride or pooling, MeteorNet needs to process all the frames in each layer and thus is not efficient. In contrast, benefiting from our point tube, the proposed PSTNet is both spatially and temporally hierarchical, and thus is efficient to model long sequences. As shown in Tab. 1, from 16 frames to 24 frames, our method achieves 1.30% improvement while MeteorNet only obtains 0.29% improvement, demonstrating our method is more effective in modeling longer sequences.
>
> $\ \ \ \ \ \ \ \ \ \ \ \ \ \ \ \ \ \ \ \ \ \ \ \ \ \ \ \ \ \ \ \ \ \ \ \ $Table 1 (Appendix Section M)
>
> $\ \ \ \ \ \ $| Method$\ \ \ \ \ \ $|  # Params | 16-frame time | 16-frame acc | 24-frame acc |
>
> $\ \ \ \ \ \ $| MeteorNet | $\ \ $17.60 M$\ $ | $\ \ \ \ \ $54.56 ms$\ \ \ $ | $\ \ \ \ \ $88.21%$\ \ \ \ \ $ | $\ \ \ \ \ $88.50%$\ \ \ \ $ |
>
> $\ \ \ \ \ \ $| PSTNet$\ \ \  \ \ \ \$  | $\ \ \ $8.26 M $\ $   | $\ \ \ \ $31.92 ms $\ \ \ $ | $\ \ \ \ \ $89.90%$\ \ \ \ \ $ | $\ \ \ \ \ $91.20%     $\ \ \ $ |
>
> - **Operators.** To the best of our knowledge, our PSTNet is the first attempt to construct convolutions on raw point cloud sequences in a unified framework. Our convolution-based network outperforms MLP-based MeteorNet on all the tasks. Moreover, as shown in Tab. 1, our PSTNet uses fewer parameters and takes less time than MeteorNet, demonstrating our convolutional operations are effective and efficient.
>
> **2. Comparison to existing static point cloud convolutional methods**
>
> Although we propose a specific lightweight implementation for the spatial convolution, we did not claim it was our key novelty or contribution. The goal of this paper is to decouple spatio-temporal information, and construct spatial and temporal hierarchy networks for point cloud sequence modeling.
>
> Compared to existing static point cloud convolutions which usually contain MLPs and other layers, our lightweight spatial convolutions contain fewer parameters and are more efficient. We replace our lightweight spatial convolution in our PSTNet with PointConv, and conduct 3D action recognition on MSR-Action3D with 16 frames.
>
> $\ \ \ \ \ \ \ \ \ \ \ \ \ \ \ \ \ $Table 2 (Appendix Section N)
>
> | Method $\ \ \ \ \ \ \ \ \ \ \ \ \ \ \ \ \ \ \ \ \ \ \ \ \ \ $| # Params | time (ms) | Acc (%)|
>
> | PSTNet (original)  $\ \ \ \ \ \ \ \ \ \ \ \$ | $\ \ \ $8.26 M$\ $ | $\ \ \ $31.92$\ \ \ \ $ | $\ $89.90$\ $ |
>
> | PSTNet (with PointConv) | $\ $ 20.46 M$\ $ | $\ $102.72$\ \ \ \ \ \$| $\ $90.24$\ $ |
>
> As shown in Tab. 2, using PointConv as spatial operations only improves performance slightly compared to our lightweight spatial convolutions, but significantly increases parameters and running time.

---

> > ### Author Response · Authors · 2020-11-21
> > **More discussions and experiments (Part 2/2)**
> >
> > **3. Effect of displacement/sharing kernel in the proposed lightweight spatial convolution.**
> >
> > The goal of displacement and sharing kernels is to generate a unique spatial kernel for each displacement so that the spatial convolution is able to capture spatial structure like conventional convolutions.  To this end, we first use a sharing kernel to increase point feature dimension to improve the feature representation ability. Then, a displacement kernel is to capture spatial local structure based on point displacements.  We conduct 3D action recognition on MSR-Action3D to study the effect of displacement/sharing kernel.
> >
> > $\ \ \ \ \ \ \ \ \ \ \ \ \ \ $Table 3 (Appendix Section L)
> >
> > | displacement | sharing | 16-frame | 24-frame |
> >
> > | $\ \ \ \ \ \ \ \ \ $✓$\ \ \ \ \ \ \ \ \ $ | $\ \ \ \ \ \ \ \ \ \ \ \ \$ | $\ \ \$58.25%$\ \$ | $\ \ $61.67%$\ \ $ |
> >
> > | $\ \ \ \ \ \ \ \ \ \ \ \ \ \ \ \ \ \ \ \ \ $ | $\ \ \ \ \ $✓$\ \ \ \ $ | $\ \ \$86.35%$\ \$ | $\ \ $87.46%$\ \ $ |
> >
> > | $\ \ \ \ \ \ \ \ \ $✓$\ \ \ \ \ \ \ \ \ $ | $\ \ \ \ \ $✓$\ \ \ \ $ | $\ \ \$89.90%$\ \$ | $\ \ $91.20%$\ \ $ |
> >
> > As shown in Tab. 3, both displacement and sharing kernel is important in our spatial convolution.
> >
> > **4. Irregular frame sampling**
> >
> >  We assume the timesteps of dynamic point clouds are regular because commercial widely-used LiDARs and depth sensors capture point cloud sequences at a fixed FPS (frames per second). For instance, the frame rates of Intel RealSense and Kinetic V2 are 30 FPS. However, our method is not restricted to this. When the frames are not irregularly sampled, we can employ frame interpolation to achieve evenly sampled frames along the temporal dimension.
> >
> > Here, we conduct 3D action recognition on MSR-Action3D with irregularly sampled frames. Clip length is originally 24 and we randomly remove 8 frames from each clip. Replication and Iterative Closest Point (ICP) are used to interpolate missing frames.
> >
> >  $\ \ \ $Table 4 (Appendix Section O)
> >
> > |Method$\ \ \ \ \ \ \ \ \ \ \ \ \ \ \ \ \ \ \ $ | Acc (%)|
> >
> > |MeteorNet$\ \ \ \ \ \ \ \ \ \ \ \ \ \ $ | $\ \ $88.40 |
> >
> > |MeteorNet (full)$\ \ \ \ \ \ $ | $\ \ $88.50 |
> >
> > |PSTNet (replication) | $\ \ $90.56 |
> >
> > |PSTNet (ICP)$\ \ \ \ \ \ \ \ \ \ \ $ | $\ \ $90.94 |
> >
> > Even with the replication interpolation, our PSTNet achieves higher performance than MeteorNet using all frames. This also indicates that PSTNet is able to model irregularly sampled dynamic point clouds.
> >
> > **5. Comparison to PST convolution without decoupling (Appendix Section L)**
> >
> > As shown in Fig. 6, the decoupled PST convolutions achieve better accuracy than the non-decoupled version. For example, with 24 frames, the decoupled PSTNet achieves 91.20% on MSR-Action, while the non-decoupled PSTNet only achieves 89.56%.
> >
> > **6. Scene flow estimation**
> >
> > To show the ability for point cloud sequence modeling, we follow MeteorNet to estimate a flow vector for every point in the last frame.  Scene flow estimation is conducted on the KITTI scene flow dataset. Point tracking is not used. For more details, please refer to  Appendix Section P.
> >
> > $\ \ \ \ \ \ \ \ \ \ \ \ $Table 5 (Appendix Section P)
> >
> > | Method$\ \ \ \ \ \ \ \ \ \ $ | #Frames | End-Point-Error |
> >
> > | FlowNet3D$\ \ \ \ \ $ | $\ \ \ \ \ \ $2$\ \ \ \ \ \ $ | $\ \ \ \ \ \ \ \ $0.287$\ \ \ \ \ \ \ \ $ |
> >
> > | MeteorNet$\ \ \ \ \ $ | $\ \ \ \ \ \ $3$\ \ \ \ \ \ $ | $\ \ \ \ \ \ \ \ $0.282$\ \ \ \ \ \ \ \ $ |
> >
> > | PSTNet (ours) | $\ \ \ \ \ \ $3$\ \ \ \ \ \ $ | $\ \ \ \ \ \ \ \ $0.278$\ \ \ \ \ \ \ \ $ |
> >
> >  As expected, our method achieves promising accuracy and outperforms the state-of-the-art.
> >
> > **7. Limitation**
> >
> > As our method only exploits local operations and does not explicitly capture global dependencies, this may limit the ability of understanding scenes or spotting periodic actions. A potential improvement is to adopt non-local techniques to enhance the feature representations in spatial and temporal dimensions.
> >
> > **8. Missing related works and minor problems**
> >
> > The minor problems and missing related works have been fixed and discussed in our revision.

---

### Decision · Program_Chairs · 2021-01-07
**Final Decision**

**Decision:**

Accept (Poster)

**Comment:**

The paper presents a construction for deep learning on point clouds that evolve over time. The key characteristics of the data are irregular sampling in the spatial domain and regular sampling in the temporal domain. The presented construction addresses both these aspects of the data. The review by R3 was negative but was addressed by the authors and R3 did not participate in the discussion. The AC supports acceptance.